# Lipid-mediated intracellular delivery of recombinant bioPROTACs for the rapid degradation of undruggable proteins

Alexander Chan[1], Rebecca M. Haley [1], Mohd Altaf Najar[2,3], David Gonzalez-Martinez[1], Lukasz J. Bugaj [1], George M. Burslem [2,3], Michael J. Mitchell [1,4] & Andrew Tsourkas [1] ✉

Recently, targeted degradation has emerged as a powerful therapeutic modality. Relying on "event-driven" pharmacology, proteolysis targeting chimeras (PROTACs) can degrade targets and are superior to conventional inhibitors against undruggable proteins. Unfortunately, PROTAC discovery is limited by warhead scarcity and laborious optimization campaigns. To address these shortcomings, analogous protein-based heterobifunctional degraders, known as bioPROTACs, have been developed. Compared to small-molecule PROTACs, bioPROTACs have higher success rates and are subject to fewer design constraints. However, the membrane impermeability of proteins severely restricts bioPROTAC deployment as a generalized therapeutic modality. Here, we present an engineered bioPROTAC template able to complex with cationic and ionizable lipids via electrostatic interactions for cytosolic delivery. When delivered by biocompatible lipid nanoparticles, these modified bioPROTACs can rapidly degrade intracellular proteins, exhibiting near-complete elimination (up to 95% clearance) of targets within hours of treatment. Our bioPROTAC format can degrade proteins localized to various subcellular compartments including the mitochondria, nucleus, cytosol, and membrane. Moreover, substrate specificity can be easily reprogrammed, allowing modular design and targeting of clinically-relevant proteins such as Ras, Jnk, and Erk. In summary, this work introduces an inexpensive, flexible, and scalable platform for efficient intracellular degradation of proteins that may elude chemical inhibition.

Many endogenous proteins have smooth surfaces that preclude modulation by conventional drugs[1–3]. The inability to develop small-molecule binders against these "undruggable" proteins remains a major hurdle in the biopharmaceutical industry and limits therapeutics development for many diseases[4,5]. Crucially, targeted inhibition of oncogenic drivers is a valuable treatment strategy in many cancers, as key molecular underpinnings leading to tumor initiation, maintenance, and metastasis have been elucidated[6,7]. Despite decades of research documenting the roles of aberrant proto-oncogenes in driving tumor growth, the current arsenal of FDA-approved inhibitors

[1]Department of Bioengineering, School of Engineering and Applied Sciences, University of Pennsylvania, Philadelphia, PA, USA. [2]Department of Biochemistry and Biophysics, Perelman School of Medicine, University of Pennsylvania, Philadelphia, PA, USA. [3]Department of Cancer Biology and Epigenetics Institute, Perelman School of Medicine, University of Pennsylvania, Philadelphia, PA, USA. [4]Penn Institute for RNA Innovation, Perelman School of Medicine, University of Pennsylvania, Philadelphia, PA, USA. ✉e-mail: atsourk@seas.upenn.edu

against high-profile targets including Ras, Myc and p53 is lacking. Even when successful molecules progress to market, their therapeutic scope is limited as highlighted by two recently-developed KRAS[G12C] inhibitors: sotorasib[8] and adagrasib[9]. While the discovery of these first KRAS inhibitors marks an important milestone in medicinal chemistry, these drugs can only target a subset of Ras variants found in a fraction of cancers[10]. Effective inhibitors of Ras and other high-priority targets are highly desirable, but discovery of lead compounds remains challenging.

Efforts to overcome intractable proteins have looked beyond direct inhibition. One such approach is small-interfering RNA (siRNA), 21-25 nucleotide oligos that silence protein expression at the transcript level. While siRNA can target any protein-coding mRNA, they come with drawbacks. Critically, the efficiency of siRNA-mediated knockdown depends on the target protein's intrinsic turnover rate, and depletion of proteins with long half-lives can lag or even outlast mRNA degradation[11,12]. Non-specific protein knockdown is also possible through imperfect complementarity with off-target mRNA[13,14], leading to potential safety concerns. To address these limitations, a fully post-translational degradation system is needed.

Recently, proteolysis targeting chimeras (PROTACs) have gained considerable attention as a drug class for their ability to degrade proteins catalytically. PROTACs are heterobifunctional molecules that simultaneously engage endogenous ubiquitin-proteasome system (UPS) machinery and proteins of interest (POI) to induce POI degradation. Structurally, PROTACs comprise three domains: a POI-binding warhead, an E3-recruiting ligand, and a chemical linker separating the two binding moieties. Since they work by inducing proximity between E3s and POIs, PROTACs can be developed from any small-molecule binder, whereas traditional inhibitors require binding in active or allosteric sites to exert their pharmacological effects. The promise of targeted protein degradation has ignited interest in PROTACs, and in the past decade, dozens of oncology-focused degraders have entered clinical trials[15]. Although they don't require binding to POI functional sites for activity, PROTACs still need high-affinity warheads for target engagement. Such ligands are difficult—if not impossible—to develop for intrinsically disordered proteins and proteins lacking hydrophobic pockets. Additionally, successful degradation necessitates the assembly of stable POI:PROTAC:E3 ternary complexes, but productive complex formation is dictated by complex interactions at the POI:E3 interface and are difficult to predict a priori[16]. Thus, screening of both linker composition and linker length is often needed to empirically optimize degrader activity. This screening process is time-consuming and can oftentimes be unfruitful in generating active degraders[17].

In a related approach, fusion of a protein-based binder to either an E3 ligase or an E3 adapter results in a recombinant "bioPROTAC". Also known as ubiquibodies[18] or AdPROMs[19,20], these biologics selectively ubiquitinate target proteins for UPS-mediated degradation. Unlike their small-molecule counterparts, bioPROTAC warheads are directly fused to E3 domains and do not depend on the recruitment of endogenous ligases. This design has demonstrated remarkable versatility being modular with respect to both the warhead and the E3 ligase. In one example, Lim et al. produced bioPROTACs that successfully degraded the same substrate with four different binding scaffolds and seven different E3 ligase adapters[21]. Importantly, bioPROTAC substrate specificity is conferred by protein scaffolds rather than small-molecule ligands. Since they operate via protein-protein interactions (PPI) spanning large contact areas, bioPROTACs can degrade targets completely inaccessible to small-molecule ligands. Also, the collection of targetable proteins is vast, benefiting from the tremendous wealth of available binding proteins. To date, numerous scaffolds including nanobodies, designed ankyrin repeat proteins (DARPins), monobodies, and affibodies have been extensively developed for exquisite specificity and nanomolar-to-picomolar affinity[22]. Taken together, bioPROTACs can be developed against any protein, greatly expanding the degradation toolbox against undruggable targets to address unmet medical needs in oncology and other therapeutic areas.

To realize the full potential of bioPROTACs, methods to deliver these macromolecules into cells are urgently needed. Currently, intracellular delivery of recombinant bioPROTACs is typically achieved by electroporation or microinjection[23,24], but these physical techniques are low-throughput, cytotoxic, and unsuitable for clinical translation[25]. Alternatively, DNA or mRNA encoding desired bioPROTACs can be transfected into cells for expression[26,27], but nucleic acid methods also face several challenges. Firstly, nucleic acids require multi-step processing including transcription, translation, and polypeptide folding prior to target degradation. In addition, stability is a concern, since mRNA molecules are highly susceptible to environmental nucleases, requiring controlled, RNase-free facilities and ultracold storage conditions to avoid degradation[28]. Finally, some therapeutically interesting bioPROTACs have been reported to express poorly as mammalian transgenes despite facile production in bacterial cultures[29]. By contrast, cytosolic protein delivery is a direct method to introduce bioactive molecules into cells for immediate target degradation. Recombinant bioPROTACs offer distinct manufacturing and storage advantages, and in some cases, protein delivery may be the only option for difficult-to-express degraders.

Common approaches for cytosolic protein delivery include cell-penetrating peptides (CPPs)[30], virus-like particles[31], inorganic nanoparticles[32,33], and supramolecular polymer assemblies[34]. However, these methods suffer from complex synthesis methods[35], endosomal entrapment[36], or potential toxicities arising from bioaccumulation of nanocarrier materials[37]. We previously demonstrated that proteins fused with anionic polypeptides (ApPs) can be complexed with cationic lipids via enforced electrostatic interactions for efficient cytosolic delivery. With this technique, we efficiently delivered inhibitory antibodies[38] and small protein scaffolds[39] into the cytosol while preserving protein function. Recently, we achieved high-efficiency delivery (90%) of ApP-tagged DARPins encapsulated within ionizable lipid nanoparticles (LNPs) for the inhibition of oncogenic Ras[40]. Critically, LNP components are generally considered safe, and LNP formulations encapsulating mRNA[41,42] and siRNA[43] have been approved by the FDA.

Herein, we develop a drug-like bioPROTAC format that can be delivered into unmodified cells for the on-demand degradation of endogenous proteins. To accomplish this goal, we first sought E3 domains that induced potent degradation when fused to the N-terminus of small protein binding scaffolds. We chose DARPins as model binding domains, owing to their excellent thermostability[44,45] and high-yield expression in E. coli cultures[46]. bioPROTAC- and target-encoding plasmids were transfected into HEK 293T (293T) cells to identify lead candidates. Once top E3-DARPin formatted bioPROTACs were identified, we explored their ability to interact with charged lipids for cytosolic delivery. To validate the degradation activity of purified proteins, we used off-the-shelf cationic Lipofectamine reagent to deliver Ras-targeting bioPROTACs into GFP-KRAS expressing reporter cells. After confirming bioactivity, we screened an LNP library to identify combinations of ionizable lipids and excipients for enhanced bioPROTAC transfection and robust degradation (Fig. 1). We characterized the degradation dose-dependence and kinetics of the lead formulation and compared its activity to bioPROTAC-encoding mRNA. We surveyed the broad applicability of our LNP-delivered bioPROTAC platform to degrade diverse proteins and proteins localized to various subcellular compartments. Finally, we show that Ras-targeting bioPROTACs can inhibit the proliferation of a KRAS-mutant pancreatic ductal adenocarcinoma (PDAC) cell line. Altogether, this work expands the functionality of existing protein binders into shelf-stable, on-demand degraders.

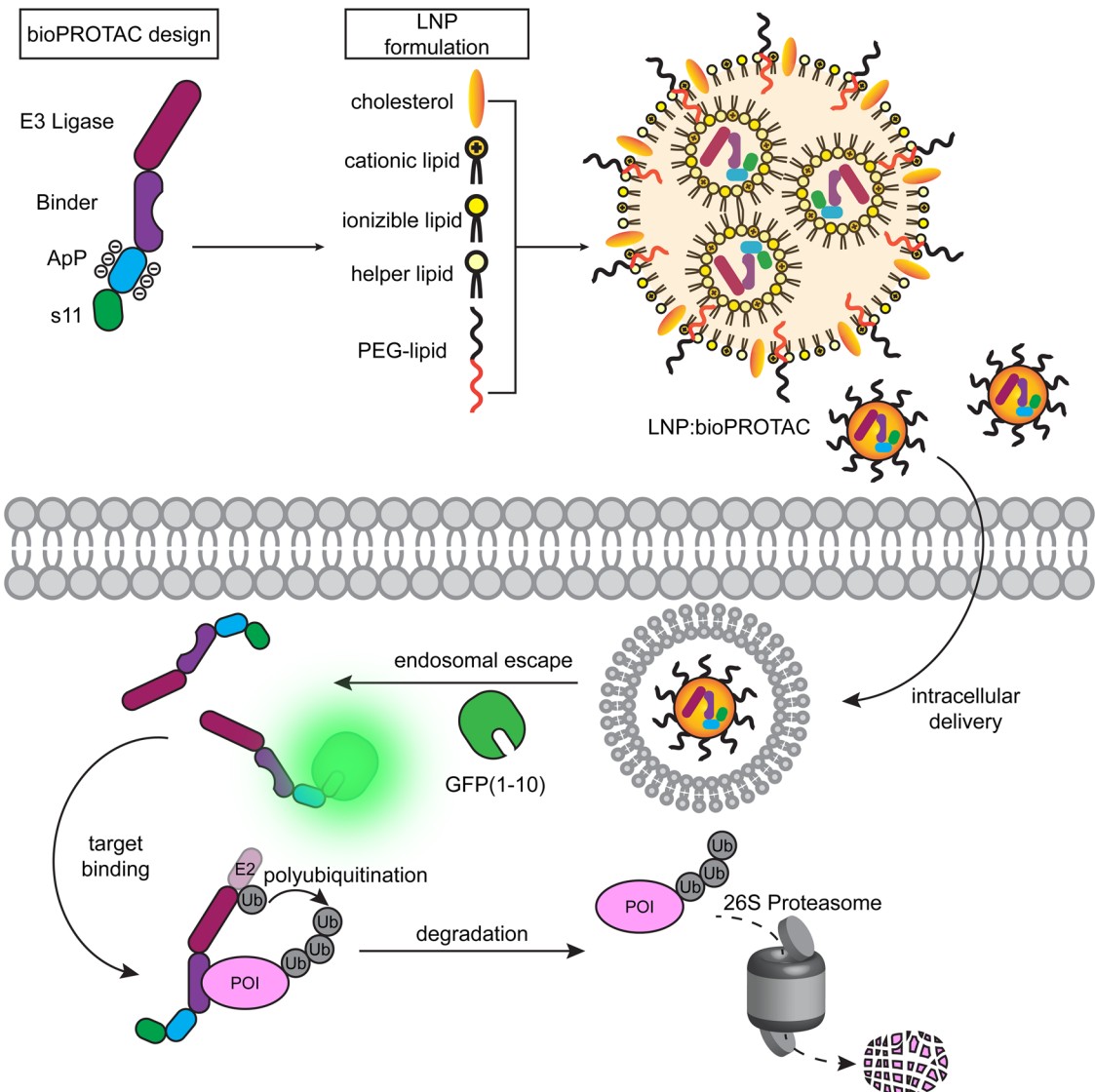

**Fig. 1 | Outline of lipid-mediated exogenous bioPROTAC delivery system.** Purified bioPROTACs include an E3 ligase, binding domain, ApP, and GFP s11 tag. The fusion proteins are formulated as LNPs including ionizable/cationic lipids, neutral helper lipids, and lipid-anchored PEG. LNP:bioPROTAC can be delivered intracellularly, and upon endosomal escape, bind to and degrade target proteins. Cytosolic protein delivery is verified by s11 complementation with GFP(1-10) expressed in reporter cell lines.

## Results

### Identifying active N-terminal E3 domains for bioPROTAC construction

A modular bioPROTAC format was established by first screening E3 ligases for degradation activity when fused to the N-terminus of a target-specific DARPin. In total, five E3 domains reported to have high activity were chosen: CHIP[18,21,47,48], SKP2[21], SOCS2[21], SPOP[21,49], IpaH9.8[50] (Fig. 2A). For all E3 proteins tested, only their degradation domains capable of recruiting E3 complex proteins or E2 conjugating enzymes were used, and the native substrate binding portions were not included in the bioPROTAC designs. Of note, the IpaH9.8 domain used in this study is not a true mammalian E3 ligase. Instead, the protein is an E3 ligase (NEL) derived from *Shigella flexneri* virulence factors, and its native role is to degrade NEMO and suppress the NF-κB inflammatory response[51,52]. Nonetheless, we included IpaH9.8 for its ability to recruit host E2 enzymes for efficient polyubiquitination. With the exception of SKP2, all E3 ligases tested have their natural substrate recognition domain at the N-terminus, and previous groups simply replaced this sequence with desired scaffolds. In contrast to this, we chose, instead, to install E3 domains at the N-terminus regardless of their natural orientation, as we previously found that C-terminal placement of polyaspartic acid ApP tags (D25, D30) preserved both expression yields and binding affinity of DARPin scaffolds[39,40]. Moreover, fusion protein designs where the ApP was placed between the binder and E3 were not considered in this screen. The GFP-binding DARPin 3G124[53] was used as the targeting domain to allow for a facile fluorescent read-out of successful degradation, and the candidate E3-DARPin formatted bioPROTACs were cloned into pcDNA vectors for characterization.

To benchmark bioPROTAC performance, 293T cells were co-transfected with pcDNA plasmids encoding GFP-KRAS and one of the five bioPROTACs at various bioPROTAC:target ratios (Fig. 2B). An additional plasmid encoding only the anti-GFP DARPin was also included as a negative control. After 48 h, cells were analyzed by flow cytometry to assess GFP degradation levels (Fig. 2C−H). No reduction in GFP was observed in cells co-transfected with 3G124 and GFP-KRAS indicating binding alone did not result in degradation. On the contrary, we saw an increase in GFP levels when 3G124 was transfected

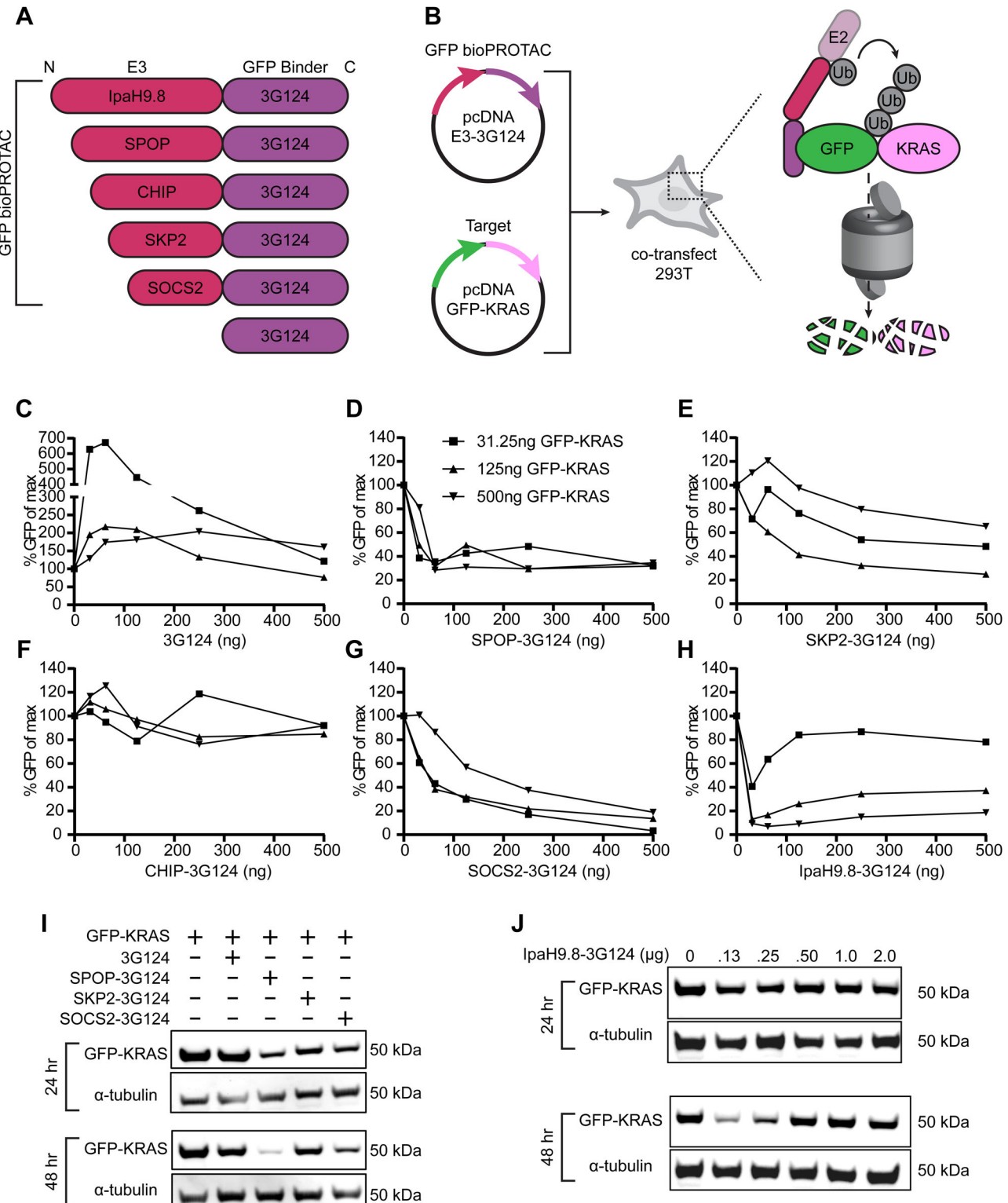

**Fig. 2 | Screening of E3 domains for bioPROTAC development. A** Protein designs for GFP-directed degradation. **B** Experimental design of co-transfection assays in 293T cells. 3G124 can bind to GFP-KRAS via the GFP handle and polyubiquitinate the target, marking it for proteasomal destruction. **C–H** Flow cytometry results of 293T cells 48 h after co-transfection with degrader- and target-encoding plasmids. Data are normalized geometric mean fluorescence intensity. No degradation was observed with the binder control (3G124) or CHIP-3G124. Both SKP2 and SOCS2

induced modest dose-dependent degradation. SPOP and IpaH9.8 transfection resulted in a dramatic reduction of target fluorescence. **I** Western blot analysis of 293T lysates following co-transfection of 2.0 μg degraders (except IpaH9.8) and 0.5 μg of GFP-KRAS. This experiment was performed twice with similar results. **J** Western blot analysis of 293T lysates following co-transfection of 0.5 μg GFP-KRAS and varying amounts of IpaH9.8-3G124. This experiment was performed twice with similar results. Source data are provided as a Source Data file.

suggesting DARPin-mediated stabilization of GFP-KRAS. Dose-dependent degradation was observed with SKP2- and SOC2-based bioPROTACs with SOCS2 exhibiting higher activity. Notably, CHIP failed to induce target elimination at any dose, despite its usage in multiple reported bioPROTAC designs. Finally, both SPOP-3G124 and IpaH9.8-3G124 exhibited the highest activity with sharp GFP reductions even at low bioPROTAC amounts. When transfected with the 500 ng of GFP-KRAS, just 62.5 ng of SPOP and IpaH9.8 bioPROTAC plasmids resulted in 70% and >90% degradation respectively. We further validated GFP-KRAS depletion by western blotting. Consistent with flow cytometry results, SPOP degraded GFP-KRAS efficiently, while SKP2 and SOCS2 displayed weaker activity (Fig. 2I). Interestingly, we observed potent IpaH9.8-induced degradation at low bioPRO-TAC:target ratios but not at high bioPROTAC levels (Fig. 2J) indicating the existence of a hook effect. This phenomenon occurs when PROTAC treatment preferentially drives the formation of POI:PROTAC and PROTAC:E3 binary complexes rather than ternary complexes needed for degradation resulting in a paradoxical reduction in degradation efficiency at higher PROTAC concentrations. As IpaH9.8 binds directly to E2 conjugating enzymes, the hook effect observed here would occur through the formation of E2:bioPROTAC binary complexes rather than through E3 saturation. Based on this screen, both SPOP and IpaH9.8 were chosen as lead E3s, and their developability was established by successful purification as DARPin-fusion proteins that retained substrate binding capabilities (Supplementary Fig. 1).

## bioPROTAC delivery with commercially available transfection reagents

Next, we replaced 3G124 with DARPinK27[54] (K27) to redirect degraders towards undruggable Ras proteins (Fig. 3A, B). As with 3G124 bio-PROTACs, transfection of 293T cells with K27-based bioPROTACs produced potent degradation of co-transfected GFP-KRAS. Here, target destruction is achieved by binding to the KRAS handle. To rule out non-specific degradation, bioPROTACs containing a null K27 mutant with abrogated binding (K27n3) were also tested and were not observed to reduce GFP levels (Fig. 3C, D). After verifying bioPROTAC functionality by transient DNA transfection, we investigated whether bioPROTACs would display similar activity as exogenously delivered purified proteins. The complete bioPROTAC format includes a target-specific scaffold fused with an N-terminal E3 domain (SPOP or IpaH9.8) and a C-terminal D25 ApP tag. A panel of control proteins was purified to dissect the necessary components of a cytosolically-delivered bio-PROTAC system (Fig. 3E, Supplementary Fig. 2b). Specifically, a binder-only control (no E3 domain), binding-deficient controls (K27n3), and non-charged (no ApP) variants were cloned and purified. For all proteins, a GFP s11 peptide was included as a reporter of intracellular delivery. The small, 16 amino acid tag does not interfere with protein function and can be used for stringent detection of cytosolic delivery when transfected into cells stably expressing the complementary GFP(1-10) fragment[55].

Either SPOP-K27-D25-s11 or IpaH9.8-K27-D25-s11 was complexed with off-the-shelf cationic Lipofectamine 2000 reagent and delivered into 293T cells stably expressing GFP-KRAS (Fig. 3F). In agreement with DNA transfection experiments, successful degradation (leftward shift in flow histograms) was observed following intracellular delivery of IpaH9.8-fused bioPROTACs (Fig. 3G). However, we did not detect GFP depletion following treatment with Lipofectamine-complexed SPOP-K27-D25-s11. It is unclear why the purified SPOP-based bioPROTAC was unable to deplete GFP-KRAS, as they were able to bind to targets (Supplementary Fig. 1c, d). Moreover, chromatograms of SPOP bio-PROTACs indicated oligomerization of the E3 ligase (Supplementary Figs. 1c, 2c-e), suggesting proper protein folding[56]. Thus, we concluded that the delivery efficiency and/or potency of SPOP-K27-D25-s11 is low and proceeded with IpaH9.8-based degraders for further development.

Next, we evaluated the roles of IpaH9.8 and ApP domains on bioPROTAC affinity, intracellular delivery, and target degradation. First, binding of the complete bioPROTAC (IpaH9.8-K27-D25-s11) against KRAS was assayed, and the affinity was found to be comparable to that of K27-s11 against KRAS (Supplementary Fig. 3). Therefore, we concluded that placement of the DARPin between E3 and ApP did not interfere with substrate recognition. To determine if IpaH9.8 and the D25 ApP domains were necessary for degrader functionality, complete Ras-targeting bioPROTACs and a series of controls were either complexed with Lipofectamine 2000 prior to delivery or directly added to 293T GFP-KRAS culture media (Fig. 4A). For each condition, 500 nM of protein was used, and degradation was quantified by flow cytometry 8 h after treatment. Cytosolic delivery of Lipofectamine-complexed IpaH9.8-K27-D25-s11 resulted in a 46% reduction of GFP-KRAS levels, whereas control proteins lacking E3, binding ability, or ApP could not deplete GFP-KRAS levels. Similarly, the complete bioPROTAC on its own produced no degradation, highlighting the membrane imper-meability of these molecules without the aid of a delivery agent.

To assay cytosolic delivery efficiency of bioPROTACs, 293T GFP(1-10) reporter cells were treated with the same panel of proteins with or without Lipofectamine (Fig. 4B, C). As expected, only proteins fused with D25 ApPs could complex with Lipofectamine and reach the cytosol. Successful cytosolic delivery was confirmed by increases both in GFP-positivity, a measure of cell transfection efficiency, and in geometric mean fluorescence intensity (MFI), a metric for the amount of protein delivered to cells. The non-binding IpaH9.8-K27n3-D25-s11 control could also be delivered into cells with the same efficiency as active bioPROTAC but was unable to degrade targets. We confirmed that this was due to the protein's inability to polyubiquitinate KRAS (Supplementary Fig. 4). Hence, we show that the E3 ligase, target-specific binder, and negatively-charged ApP are fundamental requirements for protein degradation in this bioPROTAC platform.

Since Lipofectamine is a readily available transfection reagent, we further characterized Lipofectamine-mediated bioPROTAC transfection for utility as a research tool. Following incubation with Lipo-fectamine:bioPROTAC, a dose-dependent decrease in GFP levels was observed in 293T GFP-KRAS, and at 8 h post-treatment, a maximum degradation efficiency of 45% was reached (Fig. 4D). This coincided with a peak delivery efficiency of ~22% and a 1.85-fold increase in MFI compared to untreated cells (Fig. 4E, F). Both degradation and delivery efficiencies were maximized at 125 nM bioPROTAC complexed with 2 μL of Lipofectamine. Furthermore, both degradation efficiency and delivery efficiency were found to be time dependent (Supplementary Figs. 5a–c). Lipofectamine-delivered bioPROTACs were also compared to commonly-used RNAi methods, and 293T GFP-KRAS cells were transfected with two different KRAS-targeting siRNA using Lipofecta-mine RNAiMAX (Supplementary Fig. 5d). In this head-to-head comparison, bioPROTACs displayed greater than 3 times faster degradation kinetics than siRNA (Supplementary Fig. 5e).

Taken together, purified ApP-tagged bioPROTACs could serve as a powerful research tool when paired with easily accessible cationic transfection reagents and would facilitate biological interrogation at much shorter time scales without the need to genetically modify cells. We also show that this knockdown approach circumvents one of the intrinsic limitations of siRNA (slow kinetics) for potential therapeutic applications. As another advantage over RNA, we tested the stability of purified bioPROTACs under simple storage conditions and verified that they displayed no loss in activity after four weeks when stored in PBS at 4 °C (Supplementary Fig. 6).

## Identifying an optimal LNP formulation for intracellular delivery of bioPROTAC protein

It is important to note that despite improved target knockdown compared to siRNA, complete degradation was not achieved by Lipofectamine-mediated transfection of bioPROTACs. We hypothesized

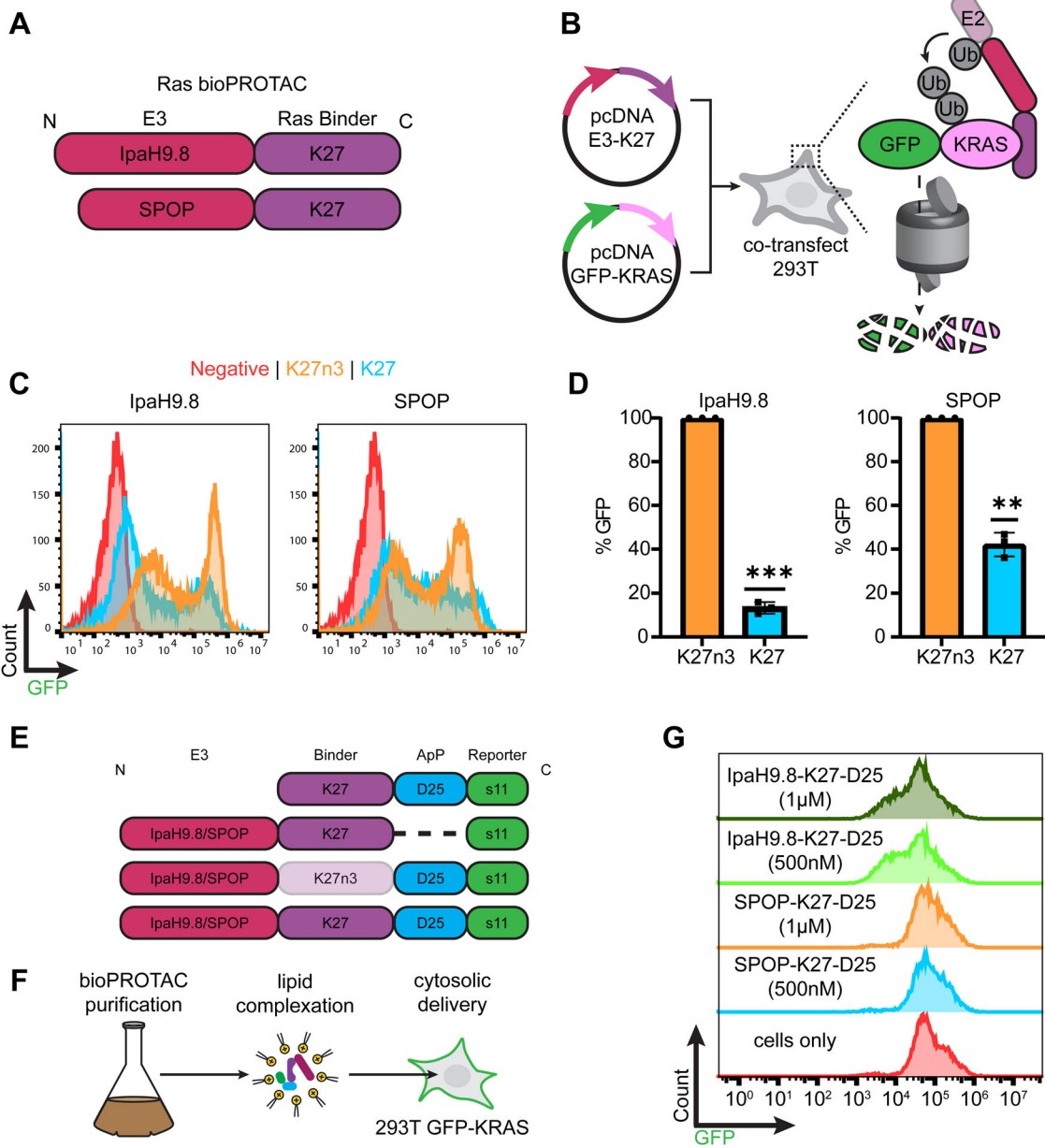

**Fig. 3 | Identification of lead E3 for bioPROTAC development. A** Either IpaH9.8 or SPOP was cloned at the N-terminus of Ras-binding K27 **B** Proposed mechanism of GFP-KRAS degradation via binding of the KRAS handle. **C** Representative flow cytometry histograms 24 h after co-transfection of the indicated bioPROTAC and GFP-KRAS. Either 2.0 μg of SPOP-K27/K27n3 plasmid or 0.25 μg of IpaH9.8-K27/K27n3 plasmid were co-transfected along with 0.5 μg of GFP-KRAS plasmid. **D** Normalization and quantitation of C. Data are mean ± SD of *n* = 3 biological replicates. A one-sample, two-tailed t test was performed. \*\**p* = 0.0029, \*\*\**p* = 0.0003. **E** Design of full bioPROTAC and negative controls. **F** Schematic of purified protein transfection using cationic Lipofectamine 2000. **G** Flow cytometry histograms of 293T GFP-KRAS cells 8 h after Lipofectamine transfection with either purified IpaH9.8-K27 or purified SPOP-K27 bioPROTACs. Source data are provided as a Source Data file.

that this incomplete elimination by Ras-targeting degraders was due to low Lipofectamine transfection efficiency. To improve intracellular protein delivery, we explored LNPs for bioPROTAC encapsulation, as they exhibit high drug loading capacity[57] and promote endosomal escape for enhanced cytosolic access[58]. To develop LNPs for bioPROTAC encapsulation, we started with three base formulations (Supplementary Table 1). Proteins were formulated into LNPs by microfluidic mixing of a bioPROTAC-containing aqueous phase and lipid-/excipient-containing ethanol phase (Fig. 5A). All three formulations performed better than Lipofectamine 2000, validating our initial reasoning for using LNPs. Excitingly, treatment with the K1 formulation led to 95% GFP-KRAS elimination at a 200 nM bioPROTAC dose (Fig. 5B, C). Interestingly, while the B6 formulation was previously optimized for DARPin-ApP delivery[40],

it only displayed a degradation efficiency of 60%. To study how LNP composition contributed to LNP:bioPROTAC formation and subsequent degradation efficiency, we generated a library of LNP formulations based on the K1 formulation which includes C12-200 as the ionizable lipid (Supplementary Table 2). Formulations incorporating 1,2-dioleoyl-sn-glycero-3-phosphoethanolamine (DOPE) as the helper lipid exhibited the highest degradation efficiency (Supplementary Fig. 7a). Ultimately, the K1 formulation was chosen as the lead LNP due to its functional potency and physical properties. The formulation retained high degradation efficiency up to one week following formulation (Supplementary Fig. 7b), had a relatively small hydrodynamic diameter of 167 nm, and exhibited near-complete encapsulation of protein cargo (Supplementary Figs. 7c–e, Supplementary Table 3).

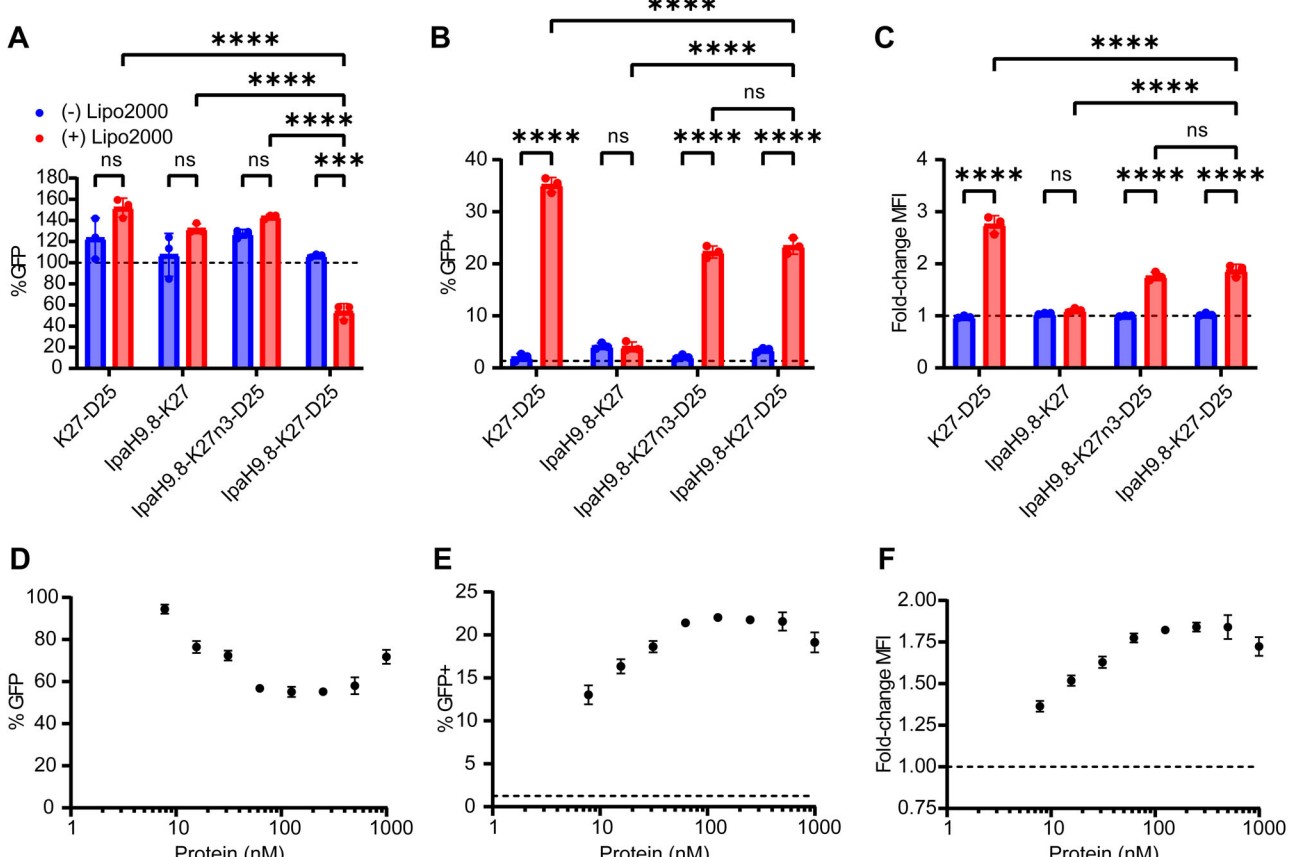

**Fig. 4 | Characterization of IpaH9.8-based bioPROTAC delivery and degradation with Lipofectamine transfection. A** Degradation efficiency of IpaH9.8-K27-D25-s11 bioPROTAC and various controls either complexed with Lipofectamine (red bars) or incubated as naked proteins (blue bars). Experiments were performed using 293T GFP-KRAS cells, and delivery was performed for 8 h. **B** Flow cytometry analysis of 293T GFP(1–10) cells for GFP-positive population following treatment with the same conditions as A. The dotted line represents the 1% threshold used to gate GFP-positive cells. **C** Fold-change in MFI of 293T GFP(1–10) cells quantified by flow cytometry with the same treatment conditions as A. and B. The dotted line represents the baseline GFP levels normalized to 1. **D** The dose-dependence of degradation efficiency on IpaH9.8-K27-D25-s11 transfection amount. For each protein dose, 2 μL Lipofectamine 2000 was used for complexation, and cells were analyzed 8 h post-delivery. **E** GFP-positive 293T GFP(1-10) cells following incubation with Lipofectamine:bioPROTAC. **F** Fold-change MFI of 293T GFP(1-10) cells following incubation with Lipofectamine:bioPROTAC. For each protein dose, 2 μL Lipofectamine 2000 was used for complexation, and cells were analyzed 8 h post-delivery. Data are mean ± SD of $n = 3$ biological replicates. For A-C, two-way ANOVA was performed followed by multiple comparisons testing. ns $p > 0.05$, ***$p \leq 0.001$, ****$p < 0.0001$. Source data are provided as a Source Data file.

The degradation dose response of K1:bioPROTAC was investigated in 293T GFP-KRAS cells by flow cytometry, and a half-maximal degradation ($DC_{50}$) of 19.5 nM was calculated at 8-h post-treatment (Fig. 5D). The acute cytotoxicity of LNP:bioPROTAC in 293T cells was determined using a lactate dehydrogenase (LDH) assay, and cell viability was found to be >90% at protein concentrations up to 100 nM (Fig. 5E). The increased target clearance by LNP:bioPROTAC was accompanied by a dramatic improvement in delivery efficiency. After treatment with 200 nM LNP-delivered bioPROTAC, 49% of 293T GFP(1–10) were GFP-positive, representing a two-fold increase in cytosolic delivery efficiency over Lipofectamine (Fig. 5F). The amount of protein delivered was commensurate, with LNP:bioPROTAC achieving a 3-fold change in MFI over the control group (Fig. 5G). For both LNP and Lipofectamine delivery approaches, we noticed that the degradation efficiency was greater than the delivery efficiency. This can be attributed to the non-stoichiometric mechanism of bioPROTACs in which one degrader molecule can catalyze the destruction of many target molecules. Moreover, the limit of detection for split GFP assays is in the low nanomolar range[36], meaning split GFP may lack sufficient sensitivity to reflect highly active degradation activity.

Next, we asked if K1:bioPROTAC could degrade endogenous Ras as opposed to GFP-KRAS fusion proteins used in characterization studies. To this end, we treated wild-type (WT) 293T cells with LNPs encapsulating either K27-D25-s11, IpaH9.8-K27-D25-s11, or IpaH9.8-K27n3-D25-s11. Lysates from treated cells were analyzed by western blotting and probed with a pan-Ras primary antibody (Fig. 5H). Closely mirroring flow cytometry results, incubation with 100 nM K1:bioPROTAC for 8 h resulted in up to 80% endogenous Ras degradation as calculated by band densitometry (Fig. 5I). In accordance with the proposed mechanism of action, neither B6:K27-D25-s11 nor K1:bioPROTAC(null) led to reductions in Ras levels owing to a lack of E3 activity and binding affinity respectively, and the bioPROTAC protein alone did not induce target degradation due its inability to enter cells without a delivery vehicle. A panel of cancer cells was also treated with LNP:bioPROTAC to validate the generalizability of our protein degrader in multiple cell types. A similar pattern of degradation was observed in HCT116 colorectal cancer cells, A549 lung cancer cells, and HT1080 fibrosarcoma cells. On average, the Ras degradation efficiency in these cell lines ranged from approximately 40% to 55% (Supplementary Fig. 8).

**Target degradation kinetics with LNP-delivered bioPROTAC**
To study the degradation kinetics of LNP-delivered bioPROTACs, we further engineered 293T GFP-KRAS cells to express iRFP-CaaX, a fluorescent membrane marker. These dual-reporter cells enabled simultaneous degradation analysis and live-cell tracking via an

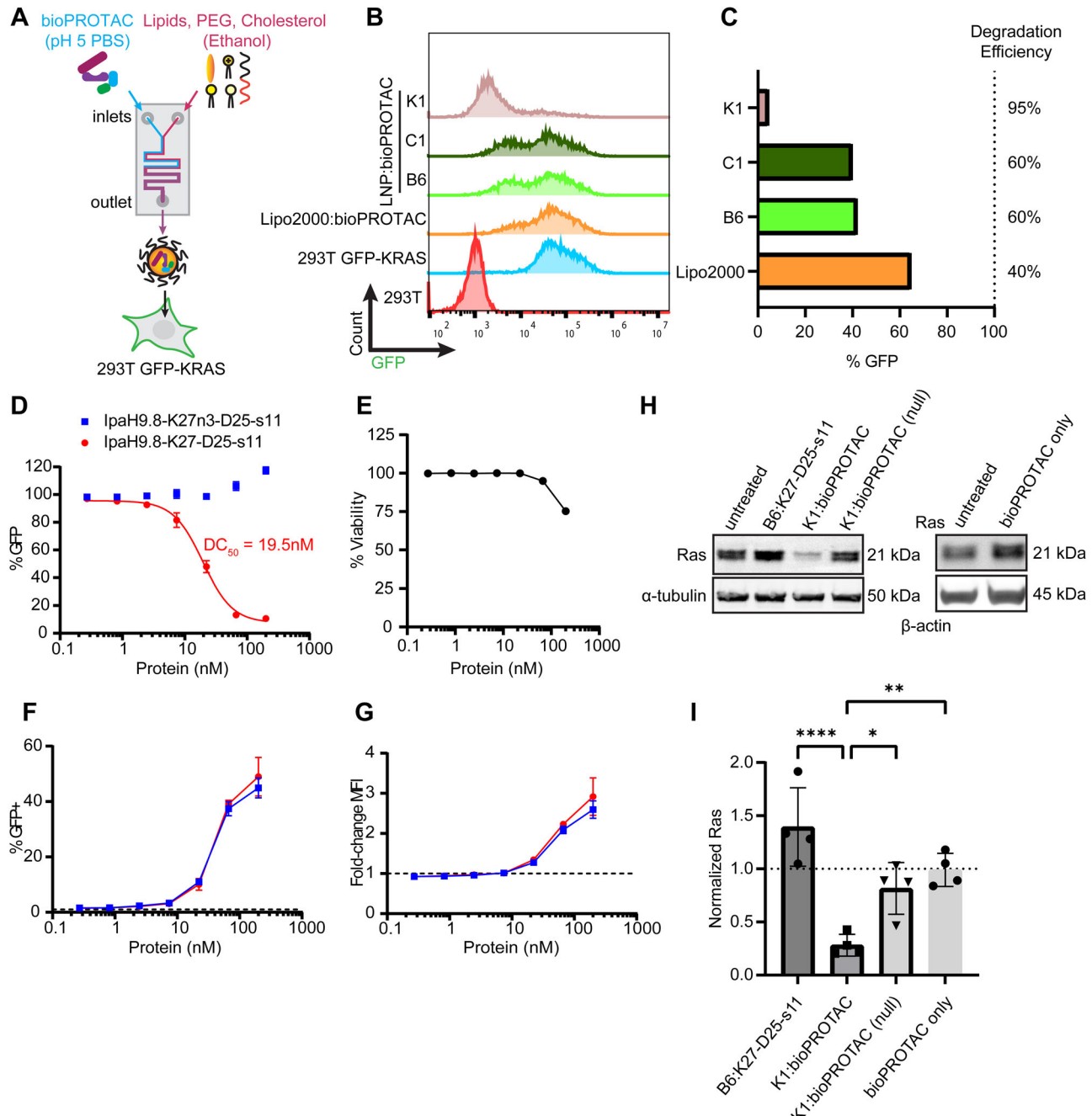

**Fig. 5 | Cytosolic bioPROTAC delivery by LNPs. A** Microfluidic mixing of an aqueous bioPROTAC solution with an ethanol solution of lipids, PEG and cholesterol was used to formulate protein LNPs. **B** Flow histograms of 293T GFP-KRAS cells treated with bioPROTACs either complexed with Lipofectamine 2000 or formulated as LNPs. **C** Quantitation of B. **D** Dose-dependent degradation in 293T GFP-KRAS with K1 LNPs encapsulating either an active bioPROTAC (red data points) or a non-binding control (blue data points). **E** The acute cytotoxicity of LNPs in 293T cells was determined by an LDH assay following treatment with the K1 formulation. **F** GFP-positive 293T GFP(1-10) cells following treatment with K1:bioPROTAC. **G** Fold-change MFI of 293T GFP(1-10) cells following treatment with K1:bioPROTAC. **H** Representative western blots of 293T lysates after cells were treated with either bioPROTAC protein only or LNPs encapsulating Ras binders, Ras degraders (bioPROTAC), or control (bioPROTAC null). **I** Quantitation of western blot degradation by band densitometry normalized to an untreated control (dotted line). For all delivery experiments, cells were incubated with proteins or LNPs for 8 h prior to analysis. Data for (**D**, **F**, **G**) are mean ± SD of $n = 3$ technical replicates. Data for (**E**) are mean of $n = 4$ technical replicates. Data for I are mean ± SD of $n = 4$ biological replicates. For (**I**), one-way ANOVA followed by multiple comparisons was performed. *$p = 0.0254$, **$p = 0.0039$, ****$p < 0.0001$. Source data are provided as a Source Data file.

orthogonal mask. As a direct comparison to direct protein delivery, we added a treatment group receiving LNP-encapsulated mRNA encoding IpaH9.8-K27 (bioPROTAC$_{mRNA}$). Cells were incubated with LNPs at dosages producing maximum degradation (Fig. 5D, Supplementary Fig. 9), and GFP was monitored for 12 h. As expected, cell-only controls exhibited no changes in GFP fluorescence intensity (Fig. 6A, D, Supplementary Movie 1). Meanwhile, both LNP-

mediated bioPROTAC protein delivery (Fig. 6B, E, Supplementary Movie 2) and bioPROTAC mRNA delivery (Fig. 6C, F, Supplementary Movie 3) resulted in rapid and robust GFP clearance, reaching maximum degradation efficiency in 5 h. The target half-life following protein delivery was calculated to be 97 min, while the half-life for mRNA was calculated to be 146 min. Treatment with either null bioPROTAC or binder only controls did not lead to degradation and

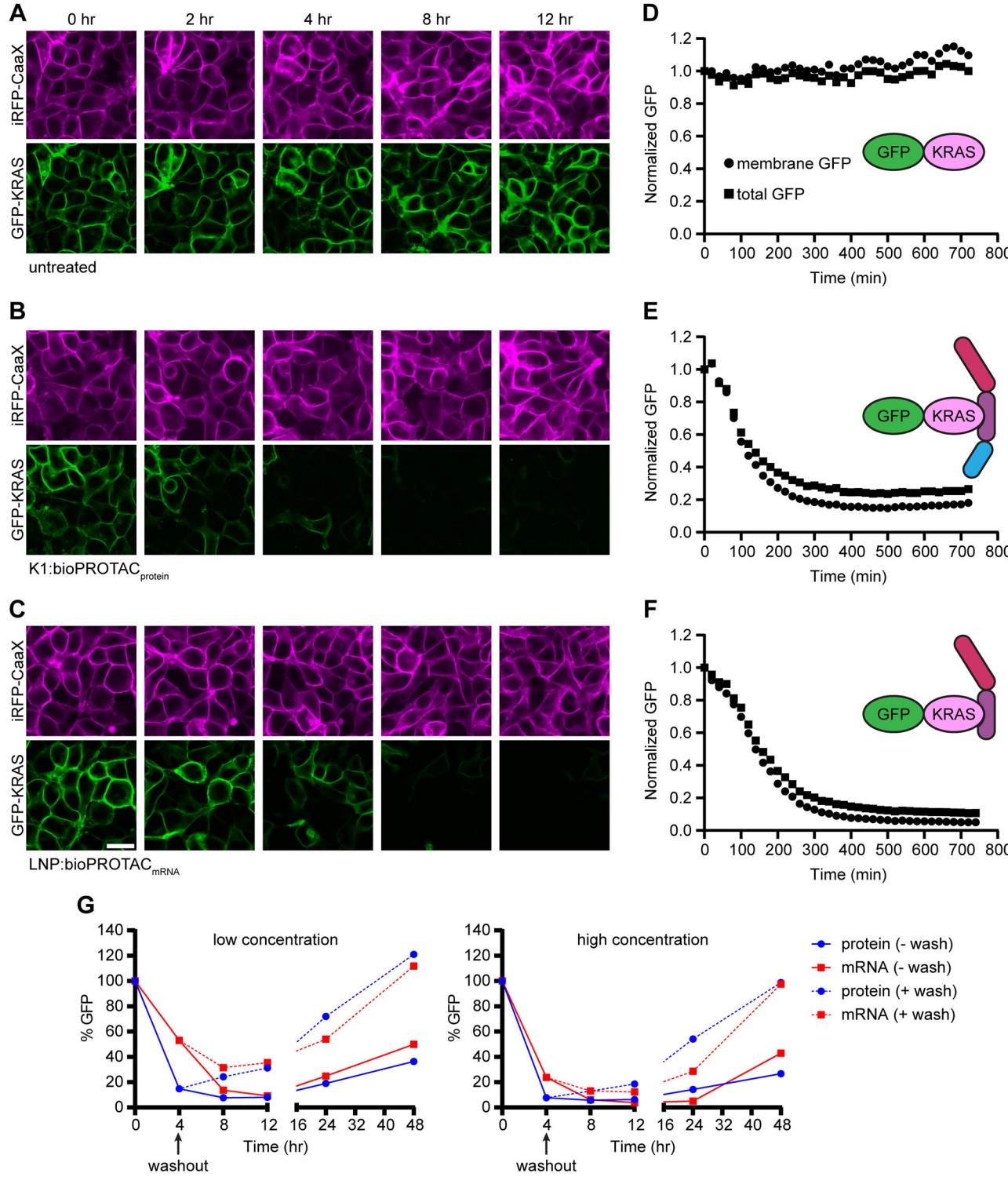

**Fig. 6 | Degradation kinetics of K1 formulation of IpaH9.8-based bioPROTACs.**
**A** Dual reporting 293T GFP-KRAS/iRFP-CaaX cells were left untreated and monitored for 12 h by fluorescence microscopy. **B** Representative fluorescent images of reporter cells treated with the K1 LNP formulation with bioPROTAC protein (200 nM) as cargo. **C** Representative fluorescent images of reporter cells treated with an LNP formulation of bioPROTAC-encoding mRNA (325 ng/mL). **D** Fluorescence intensity of individual cells from (**A**). **E** Fluorescence intensity of individual cells from (**B**). **F** Fluorescence intensity of individual cells from (**C**). Between 400-500 single cells were analyzed at each time point over the 12-h treatment window, and the mean is represented. **G** The duration of bioPROTAC-mediated degradation was determined by flow cytometry. Low dose = 50 nM protein, 100 ng/mL mRNA. High dose = 100 nM protein, 200 ng/mL mRNA. Scale bar applies to all microscopy images in (**A**–**C**) and is equal to 20 μm. Source data are provided as a Source Data file.

actually resulted in an apparent stabilization of the target protein (Supplementary Fig. 10).

For many therapeutic applications, prolonged degradation is desirable, so it is critical to understand the duration of bioPROTAC activity following LNP-mediated delivery. Flow cytometry was used to analyze LNP-treated cells at various time points up to 48 h post-delivery with either bioPROTAC$_{protein}$ or bioPROTAC$_{mRNA}$ (Fig. 6G). At both low and high doses, proteins produced sharp GFP depletion

within 4 h. The maximum degradation efficiency was nearly identical for both modalities, reaching up to ~95% target reduction by 8 h. Recovery of GFP-KRAS levels was similarly comparable between protein and RNA. Under treatment washout conditions (LNP-containing media replaced with fresh media at 4 h), complete recovery was observed by 48 h. Conversely, the sustained presence of LNPs in solution (no washout) resulted in only 30–40% recovery at 48 h.

## Global proteome response to bioPROTAC treatment

To understand how bioPROTACs were affecting protein levels globally, we performed shotgun proteomics following degrader delivery. Cells were either left untreated, treated with K1:Ras bioPROTAC protein, or transfected with LNP:bioPROTAC mRNA. We also included a group receiving K1:null bioPROTAC protein to examine the specificity of our degraders. Cells were treated for 8 h, and extracted proteins were subjected to tandem mass spectrometry (MS). Label-free quantification resulted in 4752 uniquely identified proteins common to all treatment groups. After filtering proteins with PSM > 24, a final dataset of 3827 proteins was used for downstream analysis. Proteins were considered differentially expressed if the log2 fold-change (log2fc) between treated and untreated groups was either $\leq -1$ or $\geq 1$ and the $p$-value < 0.05. In both mRNA bioPROTAC and protein bioPROTAC groups, NRAS and KRAS abundance was lower compared to no treatment, although KRAS did not meet the log2fc threshold in the protein-treated group (Fig. 7A, B). Knockdown of NRAS and KRAS was not observed in cells treated with null bioPROTAC (Fig. 7C). These results matched the expected depletion pattern in samples treated simultaneously and analyzed by western blotting (Supplementary Fig. 11a). In addition to NRAS, we also identified several proteins significantly downregulated in active bioPROTAC treatment groups but in the null bioPROTAC treatment group (Fig. 7D). Amongst these, members of the trimeric G-protein family: GNAI1, GNAI2, and GNAI3 were found to be depleted by both bioPROTAC protein and mRNA treatment. These proteins are structurally homologous to Ras-family proteins, containing P-loop and G-Box motifs also found near the DARPinK27-KRAS binding site[59]. Thus, we suspect guanine nucleotide-binding protein G(i) subunit alpha is a bona-fide off-target for DARPinK27. The relationship between the remaining proteins is less clear. However, all of them are membrane-associated. Furthermore, CHMP4A, MYO1B, and MYO1D are implicated in endosomal trafficking processes. Specifically, CHMP4A is known to facilitate lysosomal degradation of ubiquitinated membrane proteins[60], and ubiquitination-dependent lysosomal trafficking has been documented for other surface receptors[61]. These data raise the possibility that other degradation pathways including endosomal-lysosomal sorting contribute to bioPROTAC-mediated target depletion, at least in the case for membrane-associated targets.

We also noticed a high percentage of shared down proteins between mRNA and null bioPROTAC that were not identified in the protein bioPROTAC group (Fig. 7D). To better understand these downregulated proteins, we performed Gene Ontology (GO) biological process (BP) enrichment analysis and found that many members of this group mapped to RNA splicing machinery (Supplementary Fig. 11b). Prior reports have found that IpaH9.8 binds to and inhibits U2AF splicing factors[62]. In addition, 18 other U2AF-interacting proteins were downregulated, suggesting co-regulation by the IpaH9.8 domain (Supplementary Fig. 11c). Although our bioPROTAC does not include the IpaH9.8 substrate-recognition domain, U2AF binding was found—counterintuitively—to occur through the IpaH9.8 C-terminal domain[62]. Interestingly, the active bioPROTAC protein induced non-significant decreases in U2AF despite also having the IpaH9.8 catalytic module. This mitigated off-target effect could be because the protein's targeting domain redirects its activity away from U2AF. In addition, its intracellular concentration at 8 h could be lower compared to mRNA-expressed bioPROTAC.

Despite effectively degrading NRAS and KRAS, mRNA treatment led to substantially more differentially-regulated proteins compared to bioPROTAC protein delivery. Some of these proteins were assigned to mRNA processing pathways via GO enrichment analysis (Fig. 7E) and are likely cellular responses to exogenous mRNA delivery. We performed molecular function enrichment analysis on significant mRNA treatment knockdowns, and excluding Ras, we uncovered an additional 12 GTPases/GTPase-associated proteins that were downregulated by mRNA bioPROTAC. We further investigated this finding and compared the fold-change of all GTPases/GTPase-associated proteins found to be downregulated either in mRNA or protein bioPROTAC treatment groups. With the exception of RhoB, both protein and mRNA-mediated degradation of Ras and Ras-like proteins were correlated (Fig. 7F). Interestingly, we identified a subset of proteins with little homology to Ras-family proteins, including α- and β-tubulins, that were strongly downregulated by mRNA bioPROTAC but not by recombinant protein bioPROTACs, indicating plausible modality-specific degradation off-targets. Moreover, transfection with mRNA produced many additional downregulated proteins that were not accounted for by pathway or molecular function enrichment. Taken together, these results demonstrate that protein knockdown profiles can differ greatly depending on whether bioPROTACs are delivered as proteins or expressed from exogenous mRNA.

## Mechanistic exploration of bioPROTAC target- and self-degradation

Autoubiquitination and self-degradation are major concerns with our bioPROTAC design. This is because the NEL domain used in our constructs lacks the native auto-inhibitory mechanism present in full-length IpaH9.8 critical to preventing self-targeting[63]. To investigate the fate and degradation mechanism of our bioPROTACs, we produced a catalytically-dead variant by mutating the catalytic cysteine residue in IpaH9.8 NEL to alanine. We purified IpaH9.8$^{C337A}$-K27-D25-s11 and confirmed its inactivation via in vitro ubiquitination assays (Supplementary Figs. 12a–c). We found that this inactivation was not due to changes in binding affinity (Supplementary Fig. 12d). Furthermore, wild-type bioPROTACs were prone to autoubiquitination (Supplementary Fig. 12e), and this self-targeting was completely abolished in the C337A mutant. Upon delivery into GFP(1-10) reporter cells, the IpaH9.8$^{C337A}$ variant displayed higher apparent transfection efficiency compared to bioPROTACs fused to IpaH9.8$^{WT}$ (Supplementary Fig. 13a–c). This increased GFP signal indicates higher levels of bioPROTAC in the cytosol. As there is only a single amino acid difference between the two proteins, this discrepancy should not stem from differences in cargo encapsulation or delivery efficiencies. Instead, we propose that our bioPROTACs self-ubiquitinate upon intracellular delivery, reducing the cytosolic pool of degrader proteins. The addition of MG-132, a peptide proteasome inhibitor, to culture media partially rescued the degradation of GFP-KRAS and completely rescued endogenous Ras (Supplementary Figs. 13d–e). Taken together, these data indicate that bioPROTAC-mediated target degradation does occur via the ubiquitin-proteasome pathway as expected. However, the incomplete rescue of GFP-KRAS points to possible contributions from alternative degradation pathways.

To determine if the lysosome plays a role in bioPROTAC-mediated degradation, we treated cells with chloroquine and bafilomycin A1, either alone or in combination with MG-132. Like MG-132, we found that chloroquine alone partially rescued GFP-KRAS degradation following bioPROTAC (IpaH9.8$^{WT}$-K27-D25-s11) delivery. While bafilomycin A1 appears to rescue degradation of GFP-KRAS, a parallel split GFP complementation assay revealed that bioPROTAC protein delivery was almost completely suppressed (Supplementary Fig. 13f). This finding is in agreement with other reports that bafilomycin A1 prevents LNP endosomal escape by inhibiting lysosomal acidification[64,65]. On the

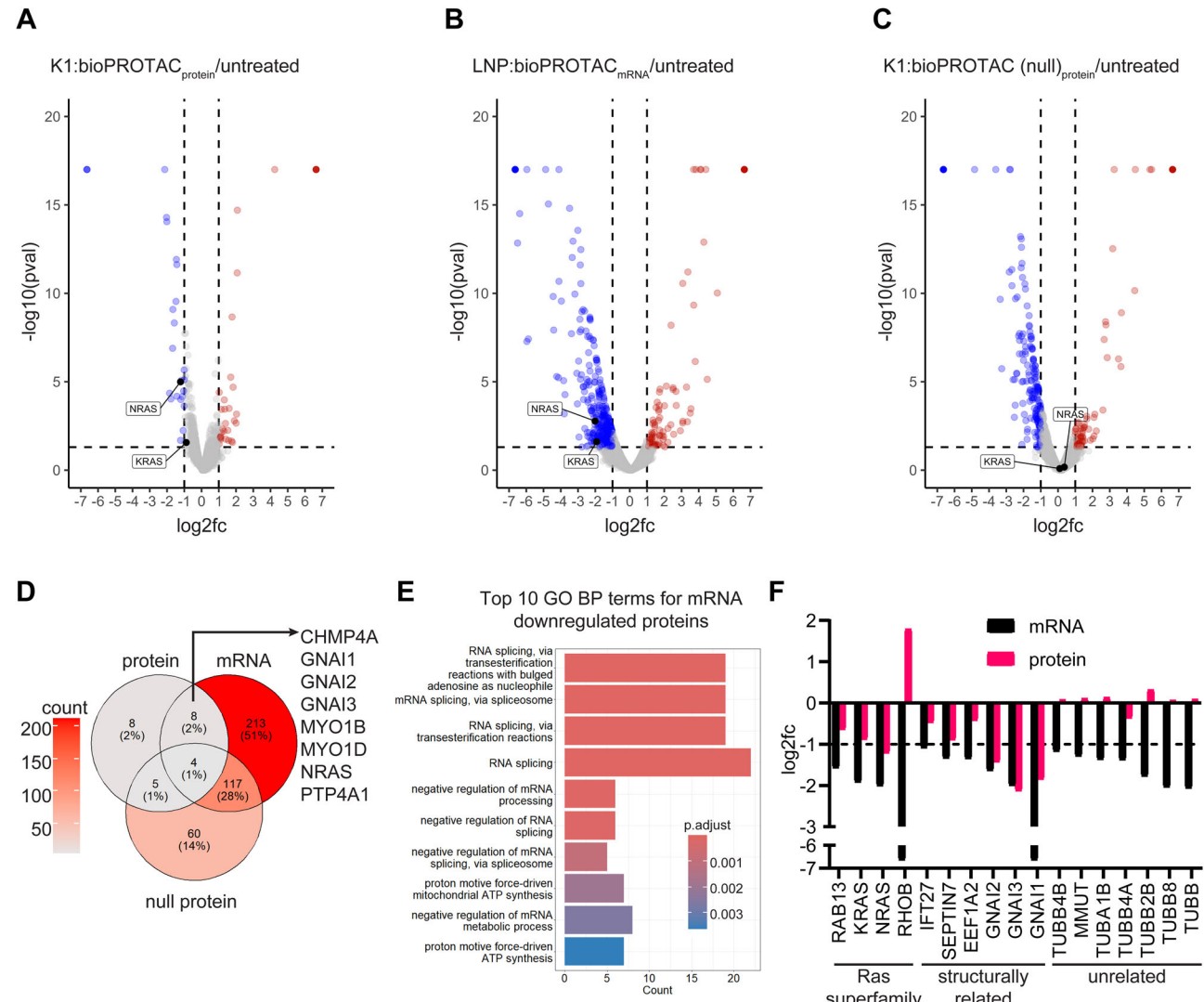

**Fig. 7 | Global profiling of 293T proteome following Ras bioPROTAC treatment.** Volcano plots display proteins identified from tandem mass spectrometry following 8-h treatment with either **A** K1-delivered Ras bioPROTAC_protein, **B** LNP-delivered bioPROTAC_mRNA, or **C** K1-delivered null bioPROTAC_protein. Upregulated and downregulated proteins are indicated as red and blue data points respectively. Both NRAS and KRAS were identified in all conditions and highlighted in volcano plots. Abundance ratio p-values were obtained by one-way ANOVA adjusted using the Benjamini-Hochberg method. **D** Venn diagram quantifying downregulated proteins from all treatment groups. The 8 downregulated proteins shared between both active bioPROTAC (protein and mRNA delivery) groups are shown. **E** GO

biological process enrichment analysis was performed on downregulated proteins unique to mRNA treatment, and the top 10 terms by adjusted p-value, as determined using the Benjamini-Hochberg method, were returned. **F** The log2fc for all GTPases/GTPase-associated proteins identified as significantly downregulated in either mRNA or protein bioPROTAC treatment are plotted. Proteins were classified based on homology to Ras-family GTPases. The dotted line marks the log2fc cutoff -1, used for identification of differentially downregulated proteins. For all treatment groups, n = 1. Log2 fold-change ratios were calculated against an untreated control group.

other hand, chloroquine did not block LNP delivery, thus implicating the lysosome in IpaH9.8 bioPROTAC-mediated degradation of GFP-KRAS. Moreover, cells treated with both MG-132 and chloroquine displayed higher split GFP complementation (6-fold MFI increase) compared to bioPROTAC delivery alone (~3-fold MFI increase), suggesting that a combination of proteasomal and lysosomal pathways partake in bioPROTAC self-destruction. Chloroquine did not rescue degradation in 293T WT cells following bioPROTAC treatment, further supporting a UPS-dominated degradation mechanism for endogenous Ras (Supplementary Fig. 13g). Taken together, these data provide evidence that multiple pathways including proteasomal and lysosomal degradation can contribute to bioPROTAC-mediated degradation. The relative contributions of each pathway are likely to be influenced by the specific bioPROTAC/target pair as well as target expression levels, among other factors.

## Modularity of our bioPROTAC format

To examine the ability of our bioPROTAC platform to degrade diverse targets, we replaced K27 with alternate DARPin sequences, redirecting their activity towards other endogenous substrates (Fig. 8A). We chose DARPins that bind extracellular signal-regulated kinase 1/2 (Erk 1/2)[66], c-Jun N-terminal kinases (Jnk)[67,68], or B-cell lymphoma-extra large (Bcl-xL)[69], as these proteins are promising therapeutic candidates being pursued for chemical inhibition. All four alternate bioPROTACs could be purified with good yield (Fig. 8B). The bioPROTACs were encapsulated in LNPs using the K1 formulation and delivered to either 293T or A549 cells. Clear depletion of Jnk and Erk bands were observed by western blotting following treatment with J1/2_2_25 and EpE89, respectively (Fig. 8C, lanes 3 and 4). By contrast J1/2_2_3 and O12_F12 failed to noticeably degrade their respective targets: Jnk and Bcl-xL (Fig. 8C, lanes 2 and 5). Notably, while both J1/2_2_25 and J1/2_2_3 are

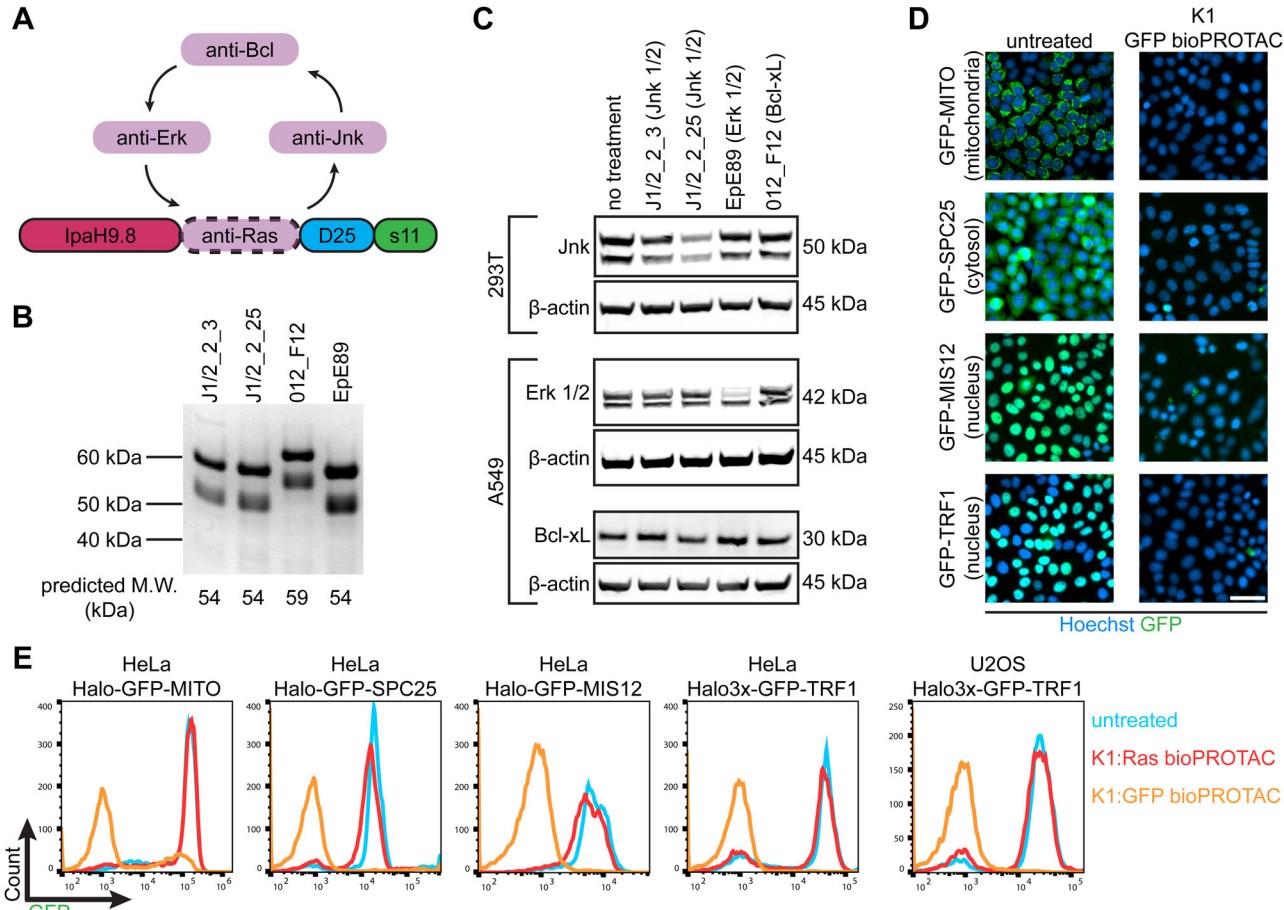

**Fig. 8 | LNP-delivered bioPROTACs are modular and widely-active. A** Schematic for "plug-and-play" design of final bioPROTAC format. **B** Four DARPins targeting three different proteins were cloned into the bioPROTAC template, purified from E. coli cultures, and analyzed by SDS-PAGE. **C** Western blot analysis reveals degradation of endogenous Jnk by K1:J1/2_2_25 bioPROTAC in 293T and degradation of Erk by K1:EpE89 bioPROTAC. No effect on Bcl-xL was observed following incubation with K1:012_F12 bioPROTAC. This experiment was performed once. **D** The anti-GFP bioPROTAC IpaH9.8-3G124-D25-s11 was formulated as K1 LNPs and delivered to

HeLa cells stably expressing GFP-fusion proteins localized to the mitochondria, cytosol, and nucleus. Degradation was analyzed by fluorescence microscopy following treatment with 100 nM protein for 8 h. This experiment was performed once. **E** Flow cytometric analysis was performed on HeLa and U2OS cells expressing various GFP-fusion proteins following treatment with 100 nM GFP bioPROTAC (K1 LNP, orange trace). To demonstrate target specificity, Ras-targeting bioPROTACs were included as a control (red trace). Scale bar applies to all microscopy images in D and is equal to 50 μm. Source data are provided as a Source Data file.

N2C-formatted DARPins identified from separate screens against Jnk, they produced different degradation outcomes. When degradation was observed, it was highly specific, as each unique bioPROTAC degraded only their intended targets while sparing the other surveyed proteins. This success rate for alternate bioPROTACs was achieved with no additional engineering of the bioPROTAC scaffold, and alternate degraders were easily produced by "plug-and-play" cloning. Thus, in agreement with previous reports of cell-expressed bioPROTACs, we conclude that our purified, ApP-tagged degrader format is modular and exhibits a high degree of design flexibility.

We next wondered if our bioPROTAC format could degrade targets localized to various intracellular compartments. Both Ras- and GFP-targeting degraders already demonstrated potent activity against GFP-KRAS and endogenous Ras, both of which associate with the inner leaflet of the plasma membrane. In addition, we tested our GFP-targeting bioPROTAC (IpaH9.8-3G124-D25-s11) against a panel of cell lines stably expressing GFP-fusions. Using the K1 formulation for cytosolic delivery, bioPROTACs effectively eliminated GFP localized to the nucleus (GFP-MIS12[70], GFP-TRF1[71]), cytosol (GFP-SPC25[70]), and mitochondria (GFP-MITO[72]). Representative fluorescent images showed near-complete degradation just 8 h after LNP incubation (Fig. 8D). We further confirmed this result by flow cytometry analysis.

Again, GFP-targeting bioPROTACs completely silenced fluorescence signal, whereas Ras-targeting bioPROTACs, used as a negative control, had no effect on target levels (Fig. 8E).

## Inhibition of pancreatic tumor cells with a Ras-degrading bioPROTAC

To demonstrate a potential therapeutic application of LNP: bioPRO-TACs, we examined the antiproliferative effects of Ras-degrading bioPROTACs when delivered into MIA PaCa-2, a PDAC line harboring a KRAS G12C driver mutation. First, we confirmed that Ras-targeting bioPROTACs were functional once delivered into MIA PaCa-2. Cells were incubated with K1 LNP formulations of either active or control bioPROTAC proteins containing either the C337A mutation and/or null K27n3 DARPin. Following 8-h protein delivery, cell lysates were analyzed by western blotting (Supplementary Fig. 14a). Following LNP treatment, Ras was depleted in a dose-dependent manner with IpaH9.8-K27-D25-s11 protein delivery, but not in any of the other treatment groups (Fig. 9A–D). We also probed phosphorylated Erk (pErk), a key effector in the canonical MAPK signaling pathway and observed its depletion in tandem with Ras degradation. Despite prior confirmation that IpaH9.8[C337A]-K27-D25-s11 retained its ability to bind Ras (Supplementary Fig. 12d), it was not able to block Ras signaling in

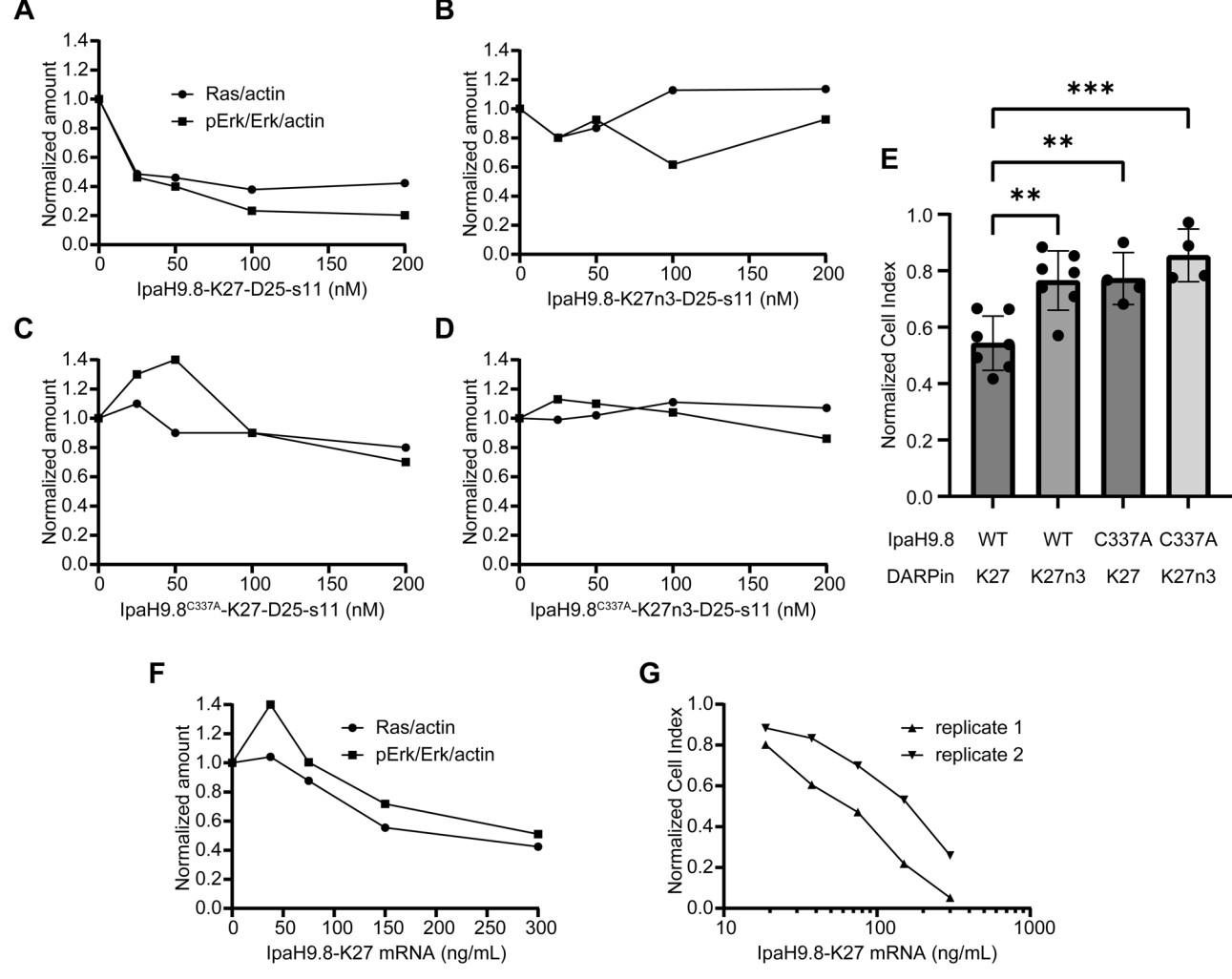

**Fig. 9 | Inhibiting proliferation of Ras-dependent pancreatic cancer cells.**
**A–D** Ras and pErk band densitometry results from MIA PaCa-2 lysates following treatment with K1:IpaH9.8-K27-D25-s11 protein (**A**), K1:IpaH9.8-K27n3-D25-s11 protein (**B**), K1:IpaH9.8$^{C337A}$-K27-D25-s11 protein (**C**), or K1:IpaH9.8$^{C337A}$-K27n3-D25-s11 protein (**D**). **E** Cell proliferation was assayed with the xCELLigence real-time cell analysis (RTCA) system, and normalized growth was calculated at 24 h post-treatment with bioPROTAC proteins formulated as K1 LNPs (56 nM dose). Data are the mean ± SD of either $n = 4$ (IpaH9.8$^{C337A}$ variants) or $n = 7$ (IpaH9.8 WT controls) biological replicates. **F** Ras and pErk band densitometry results from MIA PaCa-2

lysates following treatment with C12-200:mRNA encoding the IpaH9.8-K27 bio-PROTAC. **G** MIA PaCa-2 cells were treated with C12-200 LNPs encapsulating bio-PROTAC mRNA, proliferation was assessed by xCELLigence RTCA, and the 24-h growth was calculated. All data were normalized to untreated controls. Band densitometry was performed at 8 h post-treatment, and $n = 1$ biological replicate for each data point. A two-way ANOVA test was performed followed by multiple comparisons testing. **$p \leq 0.01$, ***$p \leq 0.001$. For (**G**), the experiment was performed twice, and the two biological replicates are shown. Source data are provided as a Source Data file.

the MIA PaCa-2 cell line (Fig. 9C, Supplementary Fig. 14a). Next, cells were treated with the same proteins formulated as K1 LNPs, and proliferation was monitored using an impedance-based confluency assay. At 24 h post-treatment, K1:IpaH9.8-K27-D25-s11 resulted in a 46% reduction in growth. These effects were statistically significant when compared to control proteins lacking binding and/or degradation ability (Fig. 9E). As the IpaH9.8-K27 bioPROTAC was the only protein able to block canonical MAPK signaling, our results indicate that anti-proliferative effects were due to Ras targeting. We compared these results against bioPROTAC mRNA treatment, which displayed comparable Ras degradation, pErk reduction, and viability inhibition at mRNA doses between 75–150 ng/mL (Fig. 9F, G).

Despite promising growth inhibition with K1:bioPROTAC, we noticed some viability loss in MIA PaCa-2 cells treated with K1-formulated null bioPROTAC at higher doses, indicating non-specific toxicity. To alleviate toxicity arising from the particles themselves, we screened variations of the K1 formulations substituting the C12-200 lipid with a series of commercially available and previously published

ionizable lipids[73] (Supplementary Fig. 15a, b). We found that LNPs incorporating SM-102 retained anti-proliferative effects, exhibiting ~70% growth inhibition while reducing non-specific toxicity to 18% at a 140 nM protein dose (Supplementary Fig. 15e). We confirmed that all bioPROTAC-containing LNPs degraded endogenous Ras (Supplementary Fig. 15i). Altogether, these results show that LNP-delivered recombinant bioPROTACs can exert therapeutic effects by degrading oncogenic proteins. By tuning LNP formulations, it is possible to obtain therapeutic profiles comparable to mRNA delivery (Supplementary Fig. 15j), a modality that has been readily adopted by the bioPROTAC field.

## Discussion
In summary, we have developed a modular recombinant bioPROTAC platform capable of on-demand, targeted protein degradation using lipid nanocarriers. The final format, incorporating an N-terminal IpaH9.8 NEL and C-terminal ApP demonstrated robust, low nanomolar activity in 7 different cell lines and could be easily reprogrammed to

polyubiquitinate diverse substrates including GFP, Ras, Erk, and Jnk. Strikingly, our bioPROTACs were active in various subcellular compartments including the cytosol, membrane, mitochondria, and nucleus. Finally, these biologic-based degraders were highly potent, exhibiting target elimination within hours of treatment. From a manufacturing standpoint, this format presents several key advantages. Firstly, our bioPROTACs are easily expressed using inexpensive bacterial cultures, enabling low-cost and accessible prototyping/testing of new degraders. Secondly, protein-based bioPROTACs exhibited no loss in activity for several weeks, are not susceptible to nuclease activity, and are amenable to typical storage conditions.

For Ras-targeting, we found that the general trend for degradation rate was, in increasing order: siRNA <DNA <mRNA <protein. While this is in line with the expected result, we note that it is difficult to make direct comparisons of degradation rates between these modalities. For example, protein expression from plasmids is heterogeneous[74], and both DNA and mRNA can generate many protein molecules per nucleic acid obfuscating direct assessments against protein delivery. Moreover, siRNA-mediated degradation is dependent on the stability of their target protein and can have vastly different depletion rates. Nonetheless, we consistently observed rapid degradation using our LNP-delivered bioPROTAC format and exhibited a target half-life of ~1.5 h following treatment. Only small-molecule systems such as DTag[75], AID[76], or conventional PROTACs[77] display faster kinetics (<30 min), but these are burdened by many challenges as previously mentioned. To further highlight differences between modalities, an apparent hook effect was only observed with pcDNA bioPROTAC transfection. This is likely due to the inclusion of a strong CMV promoter and a 5' Kozak sequence. These elements drive excessive bioPROTAC production to levels that are likely not reached by other delivery methods.

In addition to kinetic differences between degradation modalities, we also found varying target specificity depending on how the bioPROTAC was introduced. When cells were transfected with mRNA encoding a Ras-targeting bioPROTAC, we identified many non-target knockdowns via MS/MS global proteome analysis. Comparatively, delivery of the same bioPROTAC as a recombinant protein induced fewer off-target effects. While some of the additional downregulated targets in the mRNA group were identified as RNA-processing proteins, many others were not reliably explained. Taken together, our results support the idea that for some bioPROTACs, mRNA could induce more off-target degradation compared to protein delivery. Recently, pseudouridine-substituted mRNA was found to increase protein mistranslation[78], and we wonder if mutations in the DARPin variable region could generate off-target degraders from modified mRNA.

Although it has been successfully exploited as a C-terminal degradation domain, CHIP displayed no activity when fused to the N-terminus. This suggests that some E3 domains are more flexible than others for bioPROTAC development. Interestingly, we also found that IpaH9.8-K27 was better than the previously reported SPOP-K27 at eliminating Ras[79] (Fig. 3C, D). A similar result was recently published, showing that IpaH9.8 was more effective than SPOP at degrading GFP[50]. In this work, we confirm this observation, noting increased degradation rates for IpaH9.8 (1.5 h) versus SPOP (>2.5 h[79]). In addition, we found that SPOP-K27 could not be converted into an exogenously delivered degrader (Fig. 3G). We hypothesize that this failure is due to a lower intrinsic potency and/or a reduced ability to complex with lipids caused by SPOP oligomerization. Thus, we conclude that IpaH9.8 is a highly active E3 domain which can be purified from bacterial cultures and remains active following lipid-mediated cytosolic delivery. The IpaH9.8 NEL is also tolerant to positioning within chimeric proteins, as it is now confirmed to be active in both N-terminal and C-terminal designs.

Towards extending bioPROTACs for clinical applications, polymeric nanogels[80] have been harnessed for cytosolic antibody delivery and target degradation via TRIM-away[23]. This method results in 50% degradation between 4−6 h post-treatment[80]. In another approach, the ZF5.3 CPP was appended to a BCL11A-targeting, SPOP-based degrader to enhance bioPROTAC internalization[81]. With this system, 70% depletion of BCL11A was achieved within 12 h using a final protein concentration in the micromolar range. In comparison to these two recent methods, our LNP/bioPROTAC-ApP approach is both faster and more potent, with the ability to reach >90% degradation within 5−6 h using less than 100 nM of protein. This efficiency was made possible through the combination of a highly active IpaH9.8 module, a negatively charged ApP, and a compatible LNP vehicle.

From LNP screening results, we note that DOPE is a critical factor for bioPROTAC LNP performance, and we observed improved LNP stability and increased degradation in formulations incorporating DOPE as a helper lipid. This is consistent with the DOPE's known propensity to enhance endosomal escape by membrane fusion[82]. During therapeutic studies in MIA PaCa-2 PDAC cells, we noticed that some LNPs caused non-specific cytotoxicity at high doses which were alleviated by changing the ionizable lipid. This finding underscores the importance of tuning LNP properties based on cell line dependent responses. Crucially, future studies in animal models should carefully screen LNP formulations to simultaneously enhance target tissue uptake while reducing non-specific toxicity. Simultaneously, engineering of bioPROTACs for stronger degradation can lower the therapeutic dose, further reducing off-target effects.

In terms of cargo engineering, DARPins were highly tolerant as chimeric fusion proteins, highlighting their promise for targeted degradation. In addition, we were able to produce degraders from various DARPin formats including N3C (3G124, K27) and N2C (J1/2_2_3, J1/2_2_25, EpE89) scaffolds, although we failed to degrade Bcl-xL with a loop DARPin (012_F12).

Critically, we found that IpaH9.8-fused bioPROTACs were subject to autoubiquitination leading to self-destruction upon cytosolic delivery. This effect did not restrict LNP:bioPROTAC efficacy in vitro, as we still observed potent degradation in cell assays. However, autoubiquitination could limit the ability of our bioPROTAC to accumulate to therapeutic doses in vivo. One potential solution to relieve self-inhibition would be to identify lysine residues on the bioPROTAC (18 total on the Ras degrader) that are prone to self-targeting. These residues can then be selectively mutated to either alanine or arginine, rendering bioPROTACs resistant to self-degradation. Such an approach has demonstrated success in stabilizing other biodegraders[83].

Here, we used DARPins as model binders owing to their stability. Additionally, methods for screening diverse DARPins are well-established. Currently, our system relies on published small protein sequences which have been selected for a handful of intracellular targets. However, this requirement is a major limitation for bioPROTAC design against targets for which no binder has been identified. Especially for cytosolic proteins, the development of selective and high-affinity protein binders has not received the same level of attention compared to cell surface markers. If no existing binding protein exists for a target protein of interest, bioPROTAC development must be preceded by screening for target-specific binders using combinatorial scaffold libraries. Thus, we hope that bioPROTAC development can serve as a compelling motivator for the future selection of DARPins against intracellular targets. Other binding scaffolds including affibodies, nanobodies, and monobodies may also be tested in our format for even broader versatility and application scope. With the ever-expanding collection of small protein scaffolds selected for antibody-like affinity and specificity, we expect bioPROTACs will become more attractive as a therapeutic modality. The ability to deliver these protein-based degraders into cells serves as a promising avenue towards the treatment of many intractable diseases. Here, we develop such a platform, enabling cytosolic bioPROTAC protein delivery. While our results demonstrate the feasibility of this modality, the technology

is still in its early stages, and continued refinement of both the degrader and LNP carrier will be necessary for in vivo therapeutic applications.

## Methods

### Cloning

All DNA fragments used in this study were synthesized as gBlocks (IDT; Coralville, IA), and codon optimization was performed using the IDT online codon optimization tool. For mammalian cell transfection experiments, gBlocks were codon-optimized for human cell line expression, and genes were inserted into a CMV-driven pcDNA3.1 vector with a 5' Kozak sequence. Similarly, the lentivirus transfer plasmid was codon-optimized for human cell line expression. For bacterially-produced proteins, genes were codon-optimized for *E. coli* K12 expression and inserted into sortase tag-expressed protein ligation expression plasmid (pSTEPL)[84]. For cloning into pcDNA3.1 and pSTEPL, plasmids were linearized by double digestion with the appropriate restriction enzymes followed by backbone isolation using a QIAquick gel extraction kit (Qiagen; Germantown, MD). Plasmids for bacterial transformation and mammalian transfection were purified by Qiagen miniprep and Qiagen endotoxin-free maxiprep kits respectively. All plasmids were submitted for Sanger sequencing to confirm the successful cloning of the correct protein sequence.

To generate the E3-deficient binder-only control, the anti-GFP DARPin, 3G124, was cloned into the linearized pcDNA3.1 backbone between KpnI and EcoRI using In-Fusion Snap Assembly master mix (Takara Bio; San Jose, CA). To produce the bioPROTAC mammalian-expression plasmids, an SPOP$_{167-374}$-3G124 encoding gBlock was first inserted into the pcDNA3.1 backbone between KpnI and EcoRI by In-Fusion cloning. A GS-rich linker was included between SPOP and 3G124. In addition, a BamHI restriction site was designed into the gBlock between the two proteins for convenient cloning of alternate degradation domains. The remaining E3 sequences: SKP2$_{2-147}$, IpaH9.8$_{254-545}$, SOCS2$_{143-198}$, and CHIP$_{128-303}$ were inserted into this plasmid between KpnI and BamHI by In-Fusion cloning. To generate Ras-targeting bioPROTAC mammalian expression vectors, gBlocks for SPOP$_{167-374}$-DARPinK27 and SPOP$_{167-374}$-DARPinK27n3 were first cloned into pcDNA3.1 between KpnI and EcoRI. Again, both GS-rich linkers and BamHI sequences were included between the E3 domain and DARPin domain, so that IpaH9.8$_{254-545}$ could be easily substituted for SPOP. The target plasmid: pcDNA3.1 GFP-KRAS was created by cloning KRAS into an eGFP-containing plasmid between BsrGI and EcoRV.

We previously produced pSTEPL plasmids for the expression of K27-D25-s11, K27-D30-s11, K27n3-D25-s11, and K27n3-D30-s11 in *E. coli*. The DARPin domains in these plasmids were replaced with 3G124 to produce 3G124-D25-s11 and 3G124-D30-s11. To generate SPOP-3G124-D25-s11, a gBlock encoding SPOP$_{167-374}$-3G124 was inserted into pSTEPL plasmids between NdeI and XhoI to retain the C-terminal ApP and s11 sequences. Using the resulting modular template, the remaining bioPROTAC expression plasmids were produced. Specifically, the IpaH9.8$_{254-545}$ NEL sequence (wild-type or C337A mutant) was cloned between NdeI and BamHI, and the other target-specific DARPins (or null control): K27, K27n3, J1/2_2_3, J1/2_2_25, EpE89, and 012_F12 were cloned between BamHI and XhoI. To remove the D25 ApP sequence, pSTEPL plasmids were first digested with XhoI and AgeI. The backbones were then purified by gel extraction, and finally, a gBlock encoding the GFP s11 sequence was re-inserted at the same site. All proteins contained GS-rich linkers between E3, binding, ApP, and s11 reporter domains.

The GFP-KRAS lentiviral transfer plasmid was made by first linearizing the pLX304 vector (Addgene #25890) by PCR. Then, a gBlock encoding eGFP-KRAS was cloned into the pLX304 backbone using an In-Fusion HD cloning kit. The gene fragment contained a 5' Kozak sequence and a GS-rich sequence between eGFP and KRAS proteins. To

generate the iRFP-CaaX transfer plasmid, the iRFP sequence was amplified from DEST-H2B-iRFP670 (Addgene #90237) and cloned into a pHR lentiviral backbone upstream of a C-terminal CaaX sequence.

### Protein expression and purification

For all proteins, bacteria were cultured in 2YT autoinduction media including trace elements (Formedium; Norfolk, United Kingdom).

The following proteins were expressed in Shuffle T7 Express competent *E. coli* (C3029J, New England Biolabs; Ipswich, MA): SPOP-3G124-s11, SPOP-3G124-D25-s11, IpaH9.8-3G124-D25-s11, SPOP-K27-D25-s11, SPOP-K27n3-D25-s11, IpaH9.8-K27-D25-s11, IpaH9.8-K27n3-D25-s11, IpaH9.8-J1/2_2_3-D25-s11, IpaH9.8-J1/2_2_25-D25-s11, IpaH9.8-EpE89-D25-s11, and IpaH9.8-012_F12-D25-s11. All other binders and bioPROTACs were expressed in T7 Express competent *E. coli* (C2566, New England Biolabs). Additionally, T7 Express *E. coli* previously transformed with pSTEPL eGFP was used in this study for recombinant GFP expression. Expression cultures for SPOP-3G124-D25-s11, SPOP-K27-D25-s11, and SPOP-K27n3-D25-s11 were grown at 25 °C for 48 h. Expression cultures for IpaH9.8-J1/2_2_3-D25-s11, IpaH9.8-J1/2_2_25-D25-s11, IpaH9.8-EpE89-D25-s11, and IpaH9.8-012_F12-D25-s11 were grown at 30 °C for 24 h. All other proteins were expressed at 37 °C for 24 h. Cultures were grown in baffled flasks and shaken at 160–180 rpm. Using the STEPL bioconjugation/purification method, proteins were eluted from HisPur cobalt resin (Thermo Fisher Scientific) with triglycine (GGG). For some assays such as encapsulation or binding studies, dye- or biotin-labeled GGG peptides were used for C-terminal bioconjugation.

Following elution, proteins were further purified by size-exclusion chromatography using a Superdex 200 Increase 10/300 GL column (Cytiva; Marlborough, MA). Fractions were pooled and concentrated using Amicon Ultra centrifugal filters with a 10 kDa molecular weight cut-off (MilliporeSigma; Burlington, MA). The purity of the final product was characterized by SDS-PAGE, and protein concentration was determined using the BCA assay. Protein stocks were stored at −80 °C for later use.

### Cell culture

The following cell lines: 293T, 293T GFP-KRAS, 293T GFP-KRAS/iRFP-CaaX, 293T GFP(1-10), A549, and HT1080 were either obtained from our own stocks or produced for this study. HeLa cells stably expressing HaloTag-GFP-MITO, HaloTag-GFP-SPC25, HaloTag-GFP-MIS12, or HaloTag3x-GFP-TRF1 and U2OS cells stably expressing HaloTag3x-GFP-TRF1 were generously gifted by Michael Lampson. All of the above cell lines were cultured in DMEM (Thermo Fisher Scientific; Waltham, MA) supplemented with 1% pen-strep and 10% FBS. The colorectal cancer cell line, HCT116 was gifted by Michael Farwell and cultured in McCoy's 5A medium (Thermo Fisher Scientific), and the pancreatic cancer line, MIA PaCa-2 was gifted by Gregory Beatty and cultured in RPMI (Thermo Fisher Scientific). Both McCoy's 5A and RPMI media were supplemented with 1% pen-strep and 10% FBS. All cells were grown in a humidified incubator maintained at 37 °C and 5% $CO_2$ conditions. All native cell lines used in this study were authenticated by short tandem repeat (STR) profile analysis by the Penn Genomic and Sequencing Core.

### Stable cell line engineering

Lentivirus for GFP-KRAS transduction was produced using standard techniques. Briefly, equal amounts of pLX304 GFP-KRAS, pMD2.G (Addgene #12259), and psPAX2 (Addgene #12260) were transfected into 293T with Lipofectamine 2000 (Thermo Fisher Scientific) according to manufacturer's instructions. The virus-containing media was collected 48 h later, centrifuged to remove cell debris, and passed through a 0.45 μm PES syringe filter. Either 1 mL, 0.5 mL, or 100 μL of clarified lentivirus supernatant was added to separate 293T cells seeded in 6-well plates in complete DMEM. After 24 h, the culture media

was exchanged for fresh DMEM. Once cells were confluent, GFP-positive populations were isolated with the assistance of the CHOP Flow Cytometry Core using a FACSJazz sorter (BD Biosciences; Franklin Lakes, NJ).

To generate 293T GFP-KRAS/iRFP-CaaX dual reporter cells, 293T GFP-KRAS cells were first plated overnight in a 6-well plate at 300,000 cells/well. The following day, culture media was replaced with complete DMEM containing 8 μg/mL Polybrene, and 300 μL of iRFP-CaaX lentivirus was added to cells and allowed to incubate overnight. Cells were grown until confluent and sorted with the assistance of the CHOP Flow Cytometry Core using a MoFlo Astrios cell sorter (Beckman Coulter; Brea, CA) for dual GFP/RFP positivity.

## mRNA synthesis

Production of mRNA was carried out by the Penn Institute for RNA Innovation mRNA Core. The coding sequence of IpaH9.8-DARPinK27 was codon-optimized using an in-house algorithm and cloned into an in vitro transcription (IVT) vector. Using IVT, mRNA was synthesized with a co-transcriptional 5′ CleanCap and complete uridine-to-pseudouridine substitutions.

## LNP formulation

To formulate LNPs, an ethanol and an aqueous phase were mixed at a 1:3 volume ratio using a microfluidic device and pump 33 DS syringe pumps (Harvard Apparatus; Holliston, MA). To prepare the ethanol phase, ionizable lipid, 1,2-distearoyl-sn-glycero-3-phosphocholine (DSPC), 1,2-dioleoyl-sn-glycero-3-phos-phocholine (DOPC), or 1,2-dio-leoyl-sn-glycero-3-phosphoethanol-amine (DOPE), lipid-anchored polyethylene glycol (PEG) (Avanti Polar Lipids; Birmingham, AL), and cholesterol (Sigma; St. Louis, MO) components were combined. For protein encapsulating LNPs, 1,2-dioleoyl-3-trimethylammonium-pro-pane (DOTAP) was included in the ethanol phase as a fifth component, and the aqueous phase was prepared using 1x PBS, shifted to pH 5 with 150 mM sodium chloride. For mRNA encapsulating LNPs, only the original four lipid components were included in the ethanol phase, and the aqueous phase was prepared using 10 mM citrate buffer. After mixing, LNPs were subsequently dialyzed against 1x PBS for 1 h to remove ethanol.

## LNP characterization

To determine hydrodynamic radius, LNPs were diluted 100x in 1x PBS in disposable cuvettes for dynamic light scattering (DLS) measurements on the Zetasizer Nano (Malvern Instruments; Malvern, UK). LNP size (Z-average diameter) and polydispersity index (PDI) are reported as the mean ± standard deviation of $n = 3$ measurements. To quantify surface zeta potential, LNPs were diluted 100x in water in DTA1070 zeta potential cuvettes (Malvern Panalytical, Malvern, UK) for measurement on the Zetasizer Nano instrument. mRNA concentration of mRNA encapsulating LNPs was determined using A260/A280 absorbance measurements on a NanoQuant Plate (Tecan; Männedorf, Switzerland). For all LNP:protein formulations, the stated concentration refers to the total protein concentration (encapsulated and free) in solution based on the bioPROTAC input amount. Cryo-Electron Microscopy (cryo-EM) was performed by core personnel at the Beckman Center for Cryo-Electron Microscopy. LNP sample was concentrated 3-4x using Amicon Ultra Centrifugal Filters (10 kDa MWCO, Millipore Sigma). Then, 3 μL of concentrated LNPs were applied to a Quantifoil holey carbon grid which had been glow discharged. Grids were blotted, and plunge freezing was performed with liquid ethane using a Vitrobot Mark IV (Thermo Scientific). Imaging was performed on a Titan Krios (Thermo Scientific) equipped with a K3 Bioquantum.

## Protein encapsulation study

Proteins were labeled with a single C-terminal 5-carboxyte-tramethylrhodamine (5-TAMRA) dye via STEPL bioconjugation. Purified proteins were encapsulated into LNPs as described above and separated with columns packed with Sepharose CL-4B matrix (Cytiva). Fractions eluted by gravity chromatography were mixed with equal volumes of 0.1% Triton-X in black microwell plates and analyzed with a plate reader. Fractions corresponding to fluorescence peaks were pooled for DLS analysis.

## Plasmid DNA transfection

For E3 screening assays, 293T cells were plated at 100,000 cells/well in 24-well plates overnight. The following day, pcDNA plasmids for bio-PROTAC and GFP-KRAS expression were separately diluted in Opti-MEM reduced serum medium (Thermo Fisher Scientific). In total, 6 dilutions of pcDNA bioPROTAC and 4 dilutions of pcDNA GFP-KRAS were made ranging from 0 ng/well to 500 ng/well. Dilutions of the bioPROTAC and target plasmids were mixed pairwise to obtain 24 combinations of plasmids at 35 μL final volume. Each of the 24 combinations was mixed with a separate solution comprising 2.5 μL of Lipofectamine 2000 diluted into 35 μL of Opti-MEM, and the resulting mixture was incubated at RT for 20 min to promote complexation. The DNA complexes were pipetted into wells, and plates were gently shaken to disperse the solution. After 8 h, the culture media was replaced with fresh DMEM, and cells were grown for an additional 40 h. At 48 h post-transfection, cells were trypsinized and analyzed by flow cytometry.

For western blot analysis and validation of Ras-targeting bio-PROTACs, co-transfection was performed in a 6-well format with 600,000 293T cells seeded per well. In these experiments, 0.5 μg of target plasmid and 2 μg of bioPROTAC plasmid (except IpaH9.8-based degraders) were co-transfected with 5–10 μL of Lipofectamine 2000. The amount of IpaH9.8-3G124 varied for western blot analysis as indicated, and IpaH9.8-K27/K27n3 was co-transfected at 0.25 μg/well for flow cytometry. Cells were analyzed by western blotting or flow cytometry 24- or 48-h post-transfection as indicated.

## Cytosolic protein delivery

In a typical protein delivery assay, cells were seeded overnight in either 6-well or 48-well plates such that they were 70–80% confluent at the time of transfection.

Lipofectamine-mediated delivery was performed in a 48-well format. First, stock proteins were diluted with Opti-MEM to achieve 20x the specified final treatment concentration in 10 μL of Opti-MEM. Next, 2 μL of Lipofectamine 2000 was diluted into 8 μL of Opti-MEM, and the resulting mixture was thoroughly mixed with diluted protein by pipetting. The Lipofectamine/protein solution was allowed to complex for 15 min at RT, after which all 20 μL of the mixture was added to wells containing 180 μL of complete media. The corresponding lipid-free protein treatment samples were prepared by diluting stock proteins to 10x the indicated final concentration in 20 μL of Opti-MEM.

LNP-mediated cytosolic protein delivery was performed in both 6-well and 48-well formats. In either format, LNP:protein was added directly to wells to achieve the desired final concentration.

## Flow cytometry

At specified time points following either protein delivery or nucleic acid transfection, cells were detached from plates with 0.25% trypsin and pelleted at RT in a table-top centrifuge at 500 g. Cell pellets were resuspended in FACS buffer (1x PBS, 1% w/v BSA, 1 mM EDTA) and analyzed using a CytoFLEX flow cytometer (Beckman Coulter). At least 8000 cell-gated events were collected, and the geometric mean fluorescence intensity was calculated using FlowJo v10 software. For degradation assays involving 293T GFP-KRAS cells, data were normalized to untreated 293T GFP-KRAS cells and wild-type 293T cells. For split GFP delivery assays, the negative control sample was untreated GFP(1-10) cells, and the GFP-positive gate was defined such that only ~1% of the negative control sample would fall within that

positive gate. Additionally, the geometric mean fluorescence intensity for each sample was divided by the mean fluorescence intensity of the negative control, and this ratio was taken as the fold-change MFI.

## siRNA-mediated knockdown

The day before transfection, 293T GFP-KRAS cells were plated overnight at 100,000 cells/well in a 24-well plate. Custom DsiRNA (IDT) designed to target KRAS were transfected into cells at a final concentration of 10 nM using Lipofectamine RNAiMAX (Thermo Fisher Scientific). An additional group was transfected with negative control DsiRNA (IDT) at a final concentration of 10 nM. Transfections were performed according to the manufacturer's instructions. At indicated time points, cells were collected and analyzed by flow cytometry. The geometric mean fluorescence intensity was normalized to untreated and 293T wild-type samples, and the degradation rate of GFP-KRAS was estimated using a first-order decay model. The duplex sequences for siKRAS_1 are 5′-rArCrGrArUrArCrArGrCrUrArArUrUrCrArGrArArUrCrATT-3′ and 5′-rArArUrGrArUrUrCrUrGrArArUrUrArGrCrUrGrUrArUrCrGrUrCrA-3′. The duplex sequences for siKRAS_2 are 5′-rGrGrArArUrUrCrCrUrUrUrUrArUrUrGrArArArArCrArUrCAG-3′ and 5′-rCrUrGrArUrGrUrUrUrUrCrArArUrArArArArGrGrArArUrUrCrCrArU-3′.

## mRNA delivery

LNPs encapsulating bioPROTAC mRNA were added directly into cell culture wells to achieve the indicated final concentration.

## Western blotting

Equal amounts of protein were boiled in sample loading buffer (928-40004, LI-COR Biosciences; Lincoln, NE) and resolved by SDS-PAGE with 4–12% Bolt Bis-Tris polyacrylamide gels (Thermo Fisher Scientific). Proteins were transferred onto PVDF membranes, blocked with Intercept TBS blocking buffer (LI-COR), and incubated with primary antibodies (diluted according to manufacturer's recommendations) overnight at 4 °C. The primary antibodies used in this study were: mouse anti-GFP (RT0265, Bio X Cell; Lebanon, NH), rabbit anti-Ras (3965, Cell Signaling Technology; Danvers, MA), rabbit anti-SAPK/JNK (9252, Cell Signaling Technology), rabbit anti-α-tubulin (2144, Cell Signaling Technology), rabbit anti-Erk1/2 (9102, Cell Signaling Technology), mouse anti-phospho-Erk1/2 (9106, Cell Signaling Technology), rabbit anti-Bcl-xL (2764, Cell Signaling Technology), and mouse anti-β-actin (3700, Cell Signaling Technology). Following primary antibody incubation, blots were incubated goat anti-rabbit 680RD (925-68071, LI-COR) and donkey anti-mouse 800CW (925-32212, LI-COR) IR-dye functionalized secondary antibodies (1:15,000 dilution). Membranes were scanned using an Odyssey M imaging system, and relative protein abundance was calculated by band densitometry using ImageJ software.

## Binding assays

To measure the binding affinity of modified DARPins, biotinylated proteins were first produced by STEPL using a GGG-biotin peptide. Proteins were further processed as described above, serially diluted, and incubated overnight in black, 96-well streptavidin coated plates at 4 °C. The following day, 50 μL of either GFP (A42613, Thermo Fisher Scientific) or KRAS (156968, Abcam; Boston, MA) diluted to 5 μg/mL was added to wells, and plates were gently shaken for 1.5 h at RT. Then, either rabbit anti-GFP (600-401-215 L, Rockland Immunochemicals, 1:10,000 dilution; Limerick, PA) or rabbit anti-Ras (3339, Cell Signaling Technology, 1:1000 dilution) was incubated in wells for 1 h at RT. Finally, wells were incubated with a 1:4000 dilution of goat anti-Rb Ab-HRP for 1 h RT (31460, Thermo Fisher Scientific). Between each incubation step, wells were washed 3x with 200 μL of wash buffer (1x PBS, 0.05% Tween 20), and all proteins were diluted in Superblock T20 buffer (Thermo Fisher Scientific). To detect HRP, the QuantaRed substrate kit (15159, Thermo Fisher Scientific) was used according to the manufacturer's instructions. Wells not receiving GFP/KRAS were used

for background subtraction, and data were fit with a 4-parameter logistic model using GraphPad Prism v10.

## LDH cytotoxicity assay

The day before treatment, 293T cells were plated at 30,000 cells/well in a 96 well plate in 80 μL of complete media. The following day, LNPs were serially diluted in PBS to 5x the indicated final treatment concentration, and 20 μL of the diluted LNPs were added to media to achieve the specified final treatment concentration. Cells were kept in the incubator at 37 °C for 8 h before cytotoxicity was measured using the Lactate Dehydrogenase (LDH) assay kit (CK12, Dojindo Molecular Technologies; Gaithersburg, MD) according to manufacturer's instructions. Cell viability was normalized to dead and live controls in GraphPad Prism v10.

## Protein and LNP stability

For protein stability studies, purified bioPROTACs were stored at either −80 °C or 4 °C in PBS for at least 4 weeks prior to cytosolic delivery and degradation analysis. To assess LNP stability, LNP:bioPROTAC was stored at 4 °C, and degradation efficiency was tested every 2 days.

## In vitro ubiquitination assay

For each 25 μL reaction, a master mix was made by combining 2.5 μL of 10x reaction buffer (500 mM HEPES, 500 mM NaCl, 10 mM TCEP), 1 μL of ubiquitin (U-100H, R&D Systems; Minneapolis, MN), 2.5 μL of MgATP solution (B-20, R&D Systems), 0.5 μL of the human E1, UBE1 (E-304, R&D Systems), 1 μL of the human E2, UbcH5b/UBE2D2 (E2-622, R&D Systems), 3 μL of human KRAS (156968, Abcam), and 13.25 μL of MQ $H_2O$. To this master mix, 1.25 μL of either 10 μM bioPROTAC or 10 μM control bioPROTAC was added and mixed well, initiating poly-ubiquitination. An additional negative control was included where 1.25 μL of MQ $H_2O$ was added to the master mix instead of bioPROTAC. Reaction mixtures were incubated for 2 h at 37 °C and quenched with 25 μL of 2x Tricine-SDS buffer. To detect ubiquitinated KRAS, at least 0.5 μL of the sample (15 ng KRAS) was resolved by SDS-PAGE and probed by western blotting with an anti-KRAS antibody. To assay bioPROTAC autoubiquitination, proteins were first labeled with a single C-terminal TAMRA dye via STEPL, and in vitro ubiquitination was performed in the absence of the target protein. Samples were resolved by SDS-PAGE, and ubiquitin-modified bioPROTACs were visualized by fluorescence imaging.

## Sample preparation for proteomic analysis

Cells were harvested by scraping, pelleted by centrifugation, washed twice with ice-cold PBS, and snap frozen. Proteins were extracted by adding 200 μL of ice-cold lysis buffer (8 M urea, 75 mM NaCl, 50 mM Tris-HCl, pH 8.0, 1 mM EDTA, protease inhibitor cocktail) to the cell pellet followed by 3 cycles of incubation on ice (5 min) and vortexing (10 seconds). Finally, samples were sonicated with an ultrasonic homogenizer and clarified by centrifugation at 20,000 g for 10 min, and the supernatant was collected for further analysis. Protein concentration was measured using a BCA Protein Assay Kit (A53227, Thermo Fisher Scientific), and equal amounts of protein (200 μg) were taken from each sample. Samples were reduced using 5 mM dithiothreitol (DTT) at 60 °C for 30 min followed by cysteine alkylation using 20 mM iodoacetamide (IAA) for 15 min in the dark. Samples were diluted to 1 M Urea using 50 mM Tris-HCl, pH 8.0, and trypsin digestion was performed with modified sequencing-grade trypsin (Promega, Madison, WI) at a 1:30 trypsin-to-protein ratio overnight at 37 °C.

## Basic reverse phase liquid chromatographic (bRPLC) fractionation of digested peptides and C18 clean-up

Peptide fractionation was carried out using a C-18 StageTip protocol eluting with a high pH mobile phase (50 mM Triethylammonium

bicarbonate pH 8.5 (TEABC)). Briefly, the C-18 material was packed into 200 µl pipette tips, activated with 100% acetonitrile (ACN), and equilibrated with 50 mM TEABC. Peptides were resuspended in 50 mM TEABC and loaded onto the C-18 StageTips column. Samples were passed twice through the column, followed by washing with 50 mM TEABC. Finally, samples were eluted with increasing concentrations of ACN (5–50%) in TEABC to yield a total of 8 fractions and combined pairwise (1 + 5, 2 + 6, 3 + 7, 4 + 8) to give a total of 4 fractions. Peptide desalting was carried out with the C-18 StageTip method. The C-18 material was packed into 200 µl pipette tips, activated with 100% ACN, and equilibrated with 0.1% formic acid in ACN. Fractionated peptide samples were resuspended in 0.1% formic acid and loaded onto the C-18 StageTip. Samples were passed through the column twice, followed by washing with 0.1% aqueous formic acid in ACN and elution with 40% ACN in aqueous 0.1% formic acid. The eluent was dried and stored at −20 °C until LC-MS/MS analysis.

### Liquid chromatography tandem mass spectrometry (LC-MS/MS)
Cleaned peptides were analyzed on Thermo Scientific Orbitrap Exploris 240 mass spectrometer interfaced with Thermo Scientific UltiMate 3000 HPLC and UHPLC Systems. Peptide digests were reconstituted in 0.1% formic acid and were separated on an analytical column (75 µm × 15 cm) at a flow rate of 300nL/min using an increasing gradient of solvent B (0.1% formic acid in 100% acetonitrile). The total run time was set to 120 min. The mass spectrometer was operated in data-dependent acquisition mode. A survey full scan MS (m/z 400–1600) was acquired in the Orbitrap with a resolution of 6000 normalized AGC target of 300%. Data were acquired in topN with 20 dependent scans. Ions were fragmented using 37% normalized collision energy and detected at a mass resolution of 1500. Dynamic exclusion was set for 8 s with a 10ppm mass window.

### Proteomic data analysis
MS/MS searches were performed with SEQUEST against the Uniprot database for human proteins supplemented with bioPROTAC sequences using Proteome Discoverer (Version 3.0, Thermo Fisher Scientific; Bremen, Germany). The workflow included Spectrum files, Spectrum selector, SEQUEST search nodes, target decoy PSM validator, peptide validator, event detector, precursor quantifier. Data was searched in label-free quantification mode using unique peptides for quantification. Oxidation of methionine and N-terminal protein acetylation were used as dynamic modifications and carbamidomethylation of cysteine was set as a static modification. MS and MS/MS mass tolerances were set to 10 ppm and 0.05 Da, respectively. A maximum of two missed cleavage was allowed. Target-decoy database searches used for the calculation of false discovery rate (FDR) and for peptide identification FDR were set at 1%. Feature mapper and precursor ion quantifier were used for label-free quantification. Custom R scripts were used for downstream data analyses and visualizations. The mass spectrometry proteomics data have been deposited to the ProteomeXchange Consortium via the PRIDE[85] partner repository with the dataset identifier PXD049388.

### Size exclusion chromatography for protein interaction analysis
Recombinant eGFP and binding partners were diluted with PBS to a final concentration of 10 µM per protein in a final volume of 200 µL. Samples were incubated at RT for 10 min and centrifuged at 12,000 g to remove aggregates. Protein mixtures were injected into an ÄKTA pure chromatography system (Cytiva) with a Superdex 200 Increase 10/300 GL column installed. Eluted fractions were pooled and analyzed by SDS-PAGE.

### Fluorescence microscopy of HeLa and U2OS
HeLa and U2OS cells expressing GFP-fusion proteins were either left untreated or treated with 100 nM K1:bioPROTAC targeting GFP. After 7.5 h, cells were stained with Hoechst 33342 at 1 µg/mL for 20 min. After nuclei staining, media was aspirated and replaced with FluoroBrite DMEM (A1896701, Thermo Fisher Scientific). Cells were imaged using a Nikon Ti2-E microscope equipped with a Yokagawa CSU-W2 spinning disk, and an sCMOS camera (Photometrics). Images were acquired using a 20x air objective and the 405 nm and 488 nm laser lines for Hoechst and GFP respectively. Images were background subtracted and equalized with ImageJ software.

### Time-course confocal microscopy
293T GFP-KRAS/iRFP-CaaX cells were seeded overnight at 100,000 cells/well in glass 96-well plates treated with 10 µg/mL human plasma fibronectin (FC010, Thermo Fisher Scientific). The following day, LNPs were pipetted directly into wells, and cells were imaged every 20 min for 12 h using a ×40x air objective. Throughout the duration of the experiment, temperature and $CO_2$ were maintained at 37 °C and 5%, respectively, using an environmental chamber (Okolab; Sewickley, PA). Images were acquired using a Nikon Ti2-E microscope equipped with a Yokagawa CSU-W1 spinning disk, 405/488/561/640 nm laser lines, and an ORCA-fusionBT digital camera C15440 (Hamamatsu).

### Time-course image analysis
Image pre-processing was performed using ImageJ with the Morpho-LibJ plugin (https://imagej.net/plugins/morpholibj)[86]. First, GFP and RFP channel images were equalized, and GFP channel images were further background subtracted using the rolling-ball algorithm. To segment individual cells, a morphological closing operation (octagon element, radius = 15 pixels) was applied to RFP channel images, and a watershed algorithm was implemented (tolerance = 200). The resulting catchment basins were extracted and loaded into CellProfiler[87] as cell objects, while processed GFP-channel images were uploaded as grayscale images. Finally, a custom pipeline was used to measure the integrated and membrane-associated fluorescence intensity of individual cells. Between 400–500 single cells were analyzed at each time point and their calculated fluorescence intensities were averaged and normalized to fluorescence intensity at $t = 0$.

### Real-time cell viability assay
Cells were plated in a 96-well E-plate (Agilent; Santa Clara, CA) and allowed to attach overnight. The following day LNPs were added to wells to achieve the indicated final protein or mRNA concentrations, and proliferation was monitored using the xCELLigence Real-Time Cell Analysis system (Agilent). The cell index, an arbitrary unit of confluence, was normalized to untreated controls for analysis.

### Protein structural model
The structure of the Ras-targeting IpaH9.8-K27 bioPROTAC was predicted by AlphaFold[88]. The resulting model was then aligned with IpaH9.8 NEL (PDB ID: 6LOL) and DARPinK27 (PDB ID: 5O2S) with PyMol.

### Statistics
Multiple batches of proteins and LNPs were used throughout this study. Two-tailed t tests and ANOVA were used for data analysis. When appropriate, multiple comparisons testing was performed, and Bonferroni correction was applied. All statistical analyses were performed using GraphPad Prism v10.

### Materials and correspondence
Materials can be requested by academic, non-commercial institutions following a material transfer agreement. Costs associated with materials transfer including shipping will be covered by requestor. Correspondence can be directed to Andrew Tsourkas.

### Reporting summary
Further information on research design is available in the Nature Portfolio Reporting Summary linked to this article.

## Data availability

Proteomics data are available via ProteomeXchange with identifier PXD049388. Source data are provided with this paper.

## Materials availability

Materials can be requested by academic, non-commercial institutions following a material transfer agreement. Costs associated with materials transfer including shipping will be covered by requestor. Correspondence can be directed to Andrew Tsourkas.

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

## Acknowledgements

The authors thank the Penn Institute for RNA Innovation mRNA Core for mRNA production. We also thank the Children's Hospital of Philadelphia (CHOP) Flow Cytometry Core for FACS assistance. We thank the Beckman Center for Cryo-Electron Microscopy and the Institute of Structural Biology for assistance with LNP imaging. This work was supported, in part, by NIH/NCI R01 CA241661. A.C. and R.M.H. are supported by the National Science Foundation Graduate Research Fellowship (NSF-GRFP). D.G-M. and L.J.B. are supported by the NIH (R01 CA241661, R35 GM138211). M.J.M. acknowledges support from a Burroughs Wellcome Fund Career Award at the Scientific Interface (CASI), a U.S. National Institutes of Health (NIH) Director's New Innovator Award (DP2 TR002776), and an NSF CAREER Award (no. CBET-2145491). G.M.B. acknowledges support from the National Institutes of Health (R35-GM142505).

## Author contributions

A.C. and A.T. conceived the project. A.C., A.T., R.M.H., M.A.N., and G.M.B. designed experiments. A.C., R.M.H., M.A.N., and D.G-M. collected data. D.G-M. and L.J.B. gave technical advice regarding confocal microscopy experiments. A.C. created image analysis pipelines. A.C. performed data analysis and interpreted results. A.C. created figures. A.C. wrote the initial manuscript draft. A.T., M.J.M., G.M.B., and L.J.B. supervised the project. A.T., M.J.M., G.M.B., and L.J.B provided funding for the project. All authors reviewed the final manuscript.

## Competing interests

The University of Pennsylvania has filed a provisional patent application US 63/559,150 entitled "Lipid-Mediated Intracellular Delivery of Recombinant bioPROTACs for Rapid Degradation of Undruggable Proteins", with Dr. Andrew Tsourkas and Alex Chan listed as inventors. The provisional patent covers the composition of the bioPROTAC and LNP described in this manuscript and uses thereof. The University of Pennsylvania has also filed patent application WO2021077066A1 entitled "Lipid and lipid nanoparticle formulation for drug delivery", which was published on 04-22-2021, with Dr. Michael Mitchell listed as an inventor. This patent describes several of the lipids and lipid nanoparticle compositions utilized in this manuscript, namely B6, C1, C14-2, C14-4, and C14-7. The remaining authors declare no competing interest.
