## [Peer Review File · Nature Communications]

Lipid-Mediated Intracellular Delivery of Recombinant bioPROTACs for the Rapid Degradation of Undruggable Proteins

Corresponding Author: Professor Andrew Tsourkas

Figures originally included in the author's rebuttal have been redacted from this file.

Version 0:

Reviewer comments:

Reviewer #1

(Remarks to the Author)

The authors reported a recombinant bioPROTAC composed of an E3 ligase for degradation activity, DARPin for target specificity, anionic polypeptides ApPs for efficient cytosolic delivery, and an s11 peptide reporter for delivery verification. The bioPROTACs showed efficient degradation activity against undruggable KRAS proteins with a low DC50 concentration. In addition, the leverage of lipids enhanced the bioPROTACs delivery into the cytosols. Overall, the idea of lipid-mediated delivery of bioPROTACs is novel, and this work demonstrates a cost-effective and scalable platform for the intracellular degradation of undruggable proteins that resist traditional chemical inhibition.

Q1: There is no figure legend in the manuscript. Please add figure legends to clarify the information.

Q2: Is there any correlation between the ratios of GFP-KRAS and E3-3G124 in terms of degradation efficiency? Please include the rationale to choose 31.25, 125, and 500 ng GFP-KRAS in Figures 2C to 2H.

Q3: In Figure 2E, why is the degradation efficiency in the 31.25 ng GFP-KRAS group the lowest? In Figure 2H, please explain why there is an increasing trend of % GFP of max for all the groups.

Q4: Please include more data in lipid characterization, including zeta potential, TEM image, encapsulation efficiency of the bioPROTACs, and toxicity assay.

Q5: It would be hard to test the toxicity of the LNP if it's not applied in live animals. The biodistribution of the LNP is also affected by the existence of DOTAP in the formulation.

Q6: In Figure 5i, why is there a decrease in the Ras/a-tubulin ratio for the K1: null bioPROTAC group? Also, it would be better to include a pure bioPROTAC group and a B6: bioPROTAC group if comparing different LNPs.

Q7: in Figure 7, please specify the difference between J1/2_2_25 and J1/2_2_3.

Q8: If KRAS is degraded by the LNP: bioPROTAC, what about the KRAS-associated pathway inhibition, such as the RAS/MAKP pathway? Is there an evaluation of these downstream pathways?

Reviewer #2

(Remarks to the Author)

This paper describes an approach for the co-development of bioPROTACs and lipid nanocarriers to facilitate the intracellular delivery of recombinant protein bioPROTACs. This approach circumvents some key limitations of plasmid-based bioPROTAC systems, including the variability in protein expressed and the kinetic disadvantage of having to wait for

the bioPROTAC to be transcribed and translated.

To do so, they harness a charged ApP tag which facilitates loading into both commercial (lipofectamine) and custom lipid nanoparticles, and demonstrate superiority over untagged bioPROTAC variants.

Overall this is a strong manuscript with well controlled experimental design and is likely to be of interest to the scientific community. I have some minor revisions to request before publication.

This discussion of the hook effect observed with some bioPROTACs was not so clear. Since in bioPROTACs the target binder are directly fused to the E3-substrate adapter, I would not expect to see a hook effect. Are the authors suggesting that all E2s in the cell are saturated by excess bioPROTAC E2:E3 interaction? Is this something observed for all targets examined with this ligase? Some additional clarity on this would be important to potential users of this approach.

One critique of the manuscript is the strong focus on undruggable and understudied proteins in the introduction. The authors here use model protein targets which all have potent inhibitors developed already, and the requirement for a selective DARPIn or nanobody may be a challenge for other proteins which are not as well studied. Some discussion of potential limitations or cases where caution should be used in application would be welcomed in the discussion.

Reviewer #3

(Remarks to the Author)

Chan et al describe a system for the intracellular delivery of bioPROTACs as recombinant proteins, concentrating on targeting RAS. To-date approaches to deliver this type of molecule have focused on mRNA, plasmid DNA and viral methods. The ability to deliver bioPROTACs without the requirement for genetic encoding opens up experimental and therapeutic opportunities for their application. As such, it is an exciting and worthwhile area of research.

The work described in the manuscript does not provide a significant addition to the current bioPROTAC literature in terms of targets or design, applying elements of previously described tools. The observation that an IpaH9.8 bioPROTAC is the most effective in the context of this approach is interesting, but not investigated in detail. At the end of the paper the target space is expanded beyond RAS, which is tantalising, but this part of the paper is underdeveloped. The authors indicate that they have described the delivery technology before for other biologic formats, and therefore this is also not novel outside of the bioPROTAC context.

Although promising, the data presented by the authors is somewhat preliminary, focusing for the most-part on the depletion of over-expressed, as opposed to native protein. The effect on endogenous RAS levels in disease-relevant cancer cell lines was much less pronounced and confined to the supplementary data section. Given the lack of novelty for the constituent elements of the bioPROTAC system described, the study should be expanded into a more thorough analysis of the opportunity that this delivery approach affords with respect to modulating the activity of native proteins classed as “undruggable”, specifically the ability to deliver a biological effect and desired phenotype in model systems. Despite the challenges outlined by the authors, other modalities described (oligonucleotide, small molecule) have been proven in this context. Furthermore, significant work has already been done to describe the biological effect seen when using bioPROTACs against RAS, just not when delivered as a protein (work by the Rabbits and Koide groups), and as such provide a valuable benchmark.

A more specific area where the manuscript could be improved is the inclusion of orthogonal methods to validate and support the authors observations. For example, global proteomics has been applied to investigate PROTAC on- and off-target effects. The paper would benefit from such an approach, to link the loss of the target protein to a pathway effect in disease relevant cell lines, and any perturbations that might result from using a protein-based delivery system versus nucleic acid, for example. Kinetic studies could also be undertaken. Also, it wasn't clear from the text if the authors had removed native substrate binding capacity for relevant E3s, as this could have a marked effect on the specificity of these construct. In terms of target turnover and mechanism, native cell lines should also be used to confirm / explore kinetics and hook effects. Furthermore, the mechanism of action was confirmed by target ubiquitylation using a biochemical assay with purified proteins, but no data was shown in the cellular context, either by inhibiting the ability for the target to be processed through the proteasomal pathway once ubiquitylated, or removal of the bioPROTAC core activity by mutation to prevent ubiquitin addition. Also, turnover of the bioPROTAC molecule was not investigated – is the bioPROTAC degraded with the target or can it act catalytically? Is the construct turned over without target engagement? With the current format, if the bioPROTAC acts stoichiometrically this could limit the ability to generate and deliver sufficient material to provide a sustained biological effect.

Regarding the presentation of the manuscript itself, there is a lack of labelling of figures and provision of figure legends. Also, the reporting summary is brief and indicates cells lines were not validated, which is of concern. These elements need more attention.

Overall, the paper is very interesting but requires additional data to validate the approach, as outlined above. Crucially, further exploration of the biological effect observed when using these tools in relevant pre-clinical models is required– how well do they perform in this context? I expect this will be how most readers/users would want to apply this technology, so a more detailed exploration and validation is essential for maximum impact, ideally with it benchmarked against other delivery systems. Proof-of-concept for in vivo delivery would be transformative. The authors speculate on future therapeutic potential, but much work needs to be done to understand the efficacy of the constructs described, in addition to their tolerability, before this could even be considered.

Author Rebuttal letter:

School of Engineering and Applied Science
Department of Bioengineering
240 Skirkanich Hall
210 South 33rd Street
Philadelphia, PA 19104-6321
Tel 215.898.8501 Fax 215.573.3155

February 14, 2024

We thank the reviewers for the thoughtful comments and suggestions to strengthen our manuscript. We have edited our manuscript accordingly and have addressed each point below.

Reviewer #1:

The authors reported a recombinant bioPROTAC composed of an E3 ligase for degradation activity, DARPin for target specificity, anionic polypeptides ApPs for efficient cytosolic delivery, and an s11 peptide reporter for delivery verification. The bioPROTACs showed efficient degradation activity against undruggable KRAS proteins with a low DC50 concentration. In addition, the leverage of lipids enhanced the bioPROTACs delivery into the cytosols. Overall, the idea of lipid-mediated delivery of bioPROTACs is novel, and this work demonstrates a cost-effective and scalable platform for the intracellular degradation of undruggable proteins that resist traditional chemical inhibition.

Q1: There is no figure legend in the manuscript. Please add figure legends to clarify the information.

We thank the reviewer for catching this clerical error. Figure legends have been added.

Q2: Is there any correlation between the ratios of GFP-KRAS and E3-3G124 in terms of degradation efficiency? Please include the rationale to choose 31.25, 125, and 500 ng GFP-KRAS in Figures 2C to 2H.

As noted by the reviewer, an alternate way to represent the data in Figure 2 for the successful degraders is by plotting the degradation efficiency as a function of E3/target ratio (shown below). For the moderate degraders, SKP2 and SOCS2, there is a clear correlation whereby increasing the ratio of E3 to target increases degradation efficiency. There is a slight dependence on the starting amount of GFP-KRAS for these degraders (low = 31.25ng, mid = 125ng, high = 500ng). However, for the strong degraders, SPOP and IpaH9.8, this trend is not

School of Engineering and Applied Science
Department of Bioengineering
240 Skirkanich Hall
210 South 33rd Street
Philadelphia, PA 19104-6321
Tel 215.898.8501 Fax 215.573.3155

observed. We speculate that the same correlation observed for SKP2 and SOCS2 could be observed for SPOP and IpaH9.8 at much lower ratios for these two proteins owing to their potency.

[Redacted]

Secondly, our rationale for using the indicated plasmid amounts was based both on manufacturer's recommended protocol as well as empirical data. For a 24-well experiment, the suggested plasmid transfection amount is between 500ng-800ng. We chose 500ng as the upper limit of the target plasmid (GFP-KRAS). The highest amount of plasmid transfected was double this (500ng bioPROTAC + 500ng target), with all other conditions receiving \approx 1000ng DNA. Thus, the majority of the conditions tested adhered to the suggested protocol. A 4-fold serial dilution of the target was chosen to sufficiently survey the experimental space for bioPROTAC-to-target ratios. When GFP-KRAS was transfected alone (no E3) at the indicated concentrations, GFP signal spanned a large dynamic range as determined by flow cytometry,

fully capturing the extent of target expression in 293T cells (shown below). With this approach, we were able to survey over 2-logs of E3/target ratios from 0.06-16 to gain a better understanding about the degradation efficiency of our constructs.

School of Engineering and Applied Science
Department of Bioengineering
240 Skirkanich Hall
210 South 33rd Street
Philadelphia, PA 19104-6321
Tel 215.898.8501 Fax 215.573.3155

[Redacted]

Q3: In Figure 2E, why is the degradation efficiency in the 31.25 ng GFP-KRAS group the lowest? In Figure 2H, please explain why there is an increasing trend of % GFP of max for all the groups.

In figure 2E, we note the degradation efficiency in increasing order is 125ng > 31.25ng > 500ng. It is not entirely clear why the degradation efficiency is not ordered by increasing target. However, we believe that since the amount of GFP-KRAS expressed with 31.25ng plasmid is low and already close to baseline (see response to Q2), the normalized change in GFP in response to bioPROTAC co-transfection is much lower than the change at higher target levels. That is, the effect of the bioPROTAC is not sufficiently portrayed at very low target levels. Rather, at low target levels, the trend of the data is more representative of the bioPROTACs' potencies than the values themselves.

In figure 2H, we hypothesize that the increasing trend observed is due to the "hook effect". The high expression from pcDNA vectors leads to the formation of bioPROTAC:E2 and bioPROTAC:target heterodimers rather than the E2:bioPROTAC:target ternary complexes necessary for degradation. We consistently observe this effect using pcDNA vectors encoding lpaH9.8-based bioPROTACs (see Figure 2J). We also note that this is only observed with pDNA vectors and not with mRNA or protein as expression from optimized plasmids containing strong CMV promoters and 5' Kozak sequences can result in much higher internal concentrations

School of Engineering and Applied Science
Department of Bioengineering
240 Skirkanich Hall
210 South 33rd Street
Philadelphia, PA 19104-6321
Tel 215.898.8501 Fax 215.573.3155

compared to other modalities. Hook effects are commonly observed for small-molecule PROTACs. Moreover, this phenomenon has been documented with bioPROTACs as Lim et al. noted a similar effect with some of their constructs also expressed from plasmids. (see ref 21)

Q4: Please include more data in lipid characterization, including zeta potential, TEM image, encapsulation efficiency of the bioPROTACs, and toxicity assay. Additional data including zeta potential (Table S3), cryo-EM images (Figure S7C), and encapsulation efficiency (Figure S7D) were collected. An acute cytotoxicity assay for the lead LNP, K1, was included in the main text (Figure 5E).

Q5: It would be hard to test the toxicity of the LNP if it's not applied in live animals. The biodistribution of the LNP is also affected by the existence of DOTAP in the formulation.

While we agree that in vivo toxicity is an important consideration, animal experiments are outside the scope of the current work. We believe a standalone study investigating biodistribution, toxicity, uptake, degradation, and therapeutic effect should be separately examined in-depth, as nanoparticle reformulation might be necessary for animal models. We agree that DOTAP might change LNP tropism to the lung or spleen, and targeting ligands might be necessary for therapeutic targeting. However, reliable methods for generating targeted protein LNPs are not well-developed and remains an ongoing effort for our group and others. We add that we previously found that LNP formulations containing DOTAP primarily delivered to the liver (ref 40) and exhibited no toxicity.

Q6: In Figure 5i, why is there a decrease in the Ras/ α -tubulin ratio for the K1: null bioPROTAC group? Also, it would be better to include a pure bioPROTAC group and a B6: bioPROTAC group if comparing different LNPs.

While the average of this group is < 1, this is due to a single trial, whereas the other 3 trials were much closer to 1. In addition, this group is not statistically significant when compared

to untreated control, so we conclude that null DARPins did not produce a true decrease in Ras. We further thank the reviewer for a missed control. We have performed additional studies with purified protein only and included the corresponding blot and barplot in Figure 5H and 5I. However, we did not include B6:bioPROTAC, as these particles were unstable and delivered bioPROTAC sub-optimally (Table S3). Thus, we believe a better comparison between protein delivery effects is to use the LNP formulation best suited for their respective cargo.

Q7: in Figure 7, please specify the difference between J1/2_2_25 and J1/2_2_3.

School of Engineering and Applied Science
Department of Bioengineering
240 Skirkanich Hall
210 South 33rd Street
Philadelphia, PA 19104-6321
Tel 215.898.8501 Fax 215.573.3155

These are both N2C-formatted DARPins with nearly-identical structures but likely bind to different epitopes. Although this is not thoroughly explored in the original papers. We have added a clarification to the main text:

âNotably, while both J1/2_2_25 and J1/2_2_3 are N2C-formatted DARPins identified from separate screens against Jnk, they produced different degradation outcomes.â

Q8: If KRAS is degraded by the LNP: bioPROTAC, what about the KRAS-associated pathway inhibition, such as the RAS/MAKP pathway? Is there an evaluation of these downstream pathways?

We have included an additional exploration of LNP:bioPROTACs with MIA PaCa-2 PDAC cell line. We found that bioPROTACs delivered as protein or mRNA degrades Ras in a dose-dependent manner. This degradation results in a simultaneous decrease in Erk phosphorylation consistent with MAPK pathway inhibition. Finally, Ras degradation leads to anti-proliferative effects in this cell line. See newly-added Figures: 9, S14 and S15.

Reviewer #2:

This paper describes an approach for the co-development of bioPROTACs and lipid nanocarriers to facilitate the intracellular delivery of recombinant protein bioPROTACs. This approach circumvents some key limitations of plasmid-based bioPROTAC systems, including the variability in protein expressed and the kinetic disadvantage of having to wait for the bioPROTAC to be transcribed and translated.

To do so, they harness a charged ApP tag which facilitates loading into both commercial (lipofectamine) and custom lipid nanoparticles, and demonstrate superiority over untagged bioPROTAC variants.

Overall this is a strong manuscript with well controlled experimental design and is likely to be of interest to the scientific community. I have some minor revisions to request before publication.

This discussion of the hook effect observed with some bioPROTACs was not so clear. Since in bioPROTACs the target binder are directly fused to the E3-substrate adapter, I would not expect to see a hook effect. Are the authors suggesting that all E2s in the cell are saturated by excess bioPROTAC E2:E3 interaction? Is this something observed for all targets examined with this ligase? Some additional clarity on this would be important to potential users of this approach.

School of Engineering and Applied Science
Department of Bioengineering
240 Skirkanich Hall
210 South 33rd Street
Philadelphia, PA 19104-6321
Tel 215.898.8501 Fax 215.573.3155

[Redacted]

The reviewer is correct that lpaH9.8 interfaces directly with E2 ubiquitin conjugating enzymes, notably of the UBE2D family. To help explain the apparent hook effect with this construct, we cloned a C-terminal FLAG tag into our existing pcDNA lpaH9.8-K27 construct and performed western blotting on 293T-transfected cells (shown above). It is clear from the band density that lpaH9.8-K27 bioPROTAC exhibits extremely high expression at plasmid concentrations well below the recommended transfection amount. Here we transfected 0.5Åµg plasmid where the manufacture recommends a DNA amount of 4Åµg for a 6-well plate

experiment. Our blots indicate that this exogenous protein is overexpressed more than abundant and essential genes such as actin. Thus, as the reviewer points out, we posit that IpaH9.8 is saturating much of the endogenous E2 when delivered as plasmid but not as mRNA or protein.

We did not observe an apparent hook effect with the other E3 adapters (SOCS2, SPOP, SKP2), possibly because their expression level was below the threshold for system saturation at all tested plasmid amounts. In addition, SOCS2, SPOP, and SKP2 E3 adapters are components of larger E3 complexes and assemble with accessory proteins such as SKP, Cullin, and RING-box proteins prior to binding E2. On the other hand, IpaH9.8 directly binds to E2 enzymes. This may also contribute to the notable differences between IpaH9.8 and mammalian E3s tested due to differential self-regulation/kinetics of these systems.

Thus far, we have performed plasmid-based experiments targeting GFP-KRAS, and we have consistently observed this apparent hook effect with IpaH9.8. Overall, these observations
School of Engineering and Applied Science
Department of Bioengineering
240 Skirkanich Hall
210 South 33rd Street
Philadelphia, PA 19104-6321
Tel 215.898.8501 Fax 215.573.3155

of an apparent hook effect in some bioPROTAC plasmids underscore the unpredictability of plasmid-based bioPROTAC delivery.

We have added two more discussion on this topic in the main text to help clarify this point:

1. As IpaH9.8 binds directly to E2 conjugating enzymes, the hook effect observed here would occur through the formation of E2:bioPROTAC binary complexes rather than through E3 saturation

2. To further highlight differences between modalities, an apparent hook effect was only observed with pcDNA bioPROTAC transfection. This is likely due to the inclusion of a strong CMV promoter and a 5' Kozak sequence. These elements drive excessive bioPROTAC production to levels that are likely not reached by other delivery methods.

One critique of the manuscript is the strong focus on undruggable and understudied proteins in the introduction. The authors here use model protein targets which all have potent inhibitors developed already, and the requirement for a selective DARPIn or nanobody may be a challenge for other proteins which are not as well studied. Some discussion of potential limitations or cases where caution should be used in application would be welcomed in the discussion.

We agree with the reviewer and have added the following discussion in the main text:
Currently, our system relies on published small protein sequences which have been selected for a handful of intracellular targets. However, this requirement is a major limitation for bioPROTAC design against targets for which no binder has been identified. Especially for cytosolic proteins, development of selective and high affinity protein binders has not received the same level of attention compared to cell surface markers. If no existing binding protein exists for a target protein of interest, bioPROTAC development must be preceded by screening for target-specific binders using scaffold libraries. Thus, we hope that bioPROTAC development can serve as a compelling motivator for future selection of DARPIns against intracellular targets.

Reviewer #3:

Chan et al describe a system for the intracellular delivery of bioPROTACs as recombinant proteins, concentrating on targeting RAS. To-date approaches to deliver this type of molecule have focused on mRNA, plasmid DNA and viral methods. The ability to deliver bioPROTACs without the requirement for genetic encoding opens up experimental and therapeutic opportunities for their application. As such, it is an exciting and worthwhile area of research.
School of Engineering and Applied Science
Department of Bioengineering
240 Skirkanich Hall
210 South 33rd Street
Philadelphia, PA 19104-6321
Tel 215.898.8501 Fax 215.573.3155

The work described in the manuscript does not provide a significant addition to the current bioPROTAC literature in terms of targets or design, applying elements of previously described tools. The observation that an IpaH9.8 bioPROTAC is the most effective in the context of this approach is interesting, but not investigated in detail. At the end of the paper the target space is expanded beyond RAS, which is tantalising, but this part of the paper is underdeveloped. The

authors indicate that they have described the delivery technology before for other biologic formats, and therefore this is also not novel outside of the bioPROTAC context.

Although promising, the data presented by the authors is somewhat preliminary, focusing for the most-part on the depletion of over-expressed, as opposed to native protein. The effect on endogenous RAS levels in disease-relevant cancer cell lines was much less pronounced and confined to the supplementary data section.

Given the lack of novelty for the constituent elements of the bioPROTAC system described, the study should be expanded into a more thorough analysis of the opportunity that this delivery approach affords with respect to modulating the activity of native proteins classed as *undruggable*, specifically the ability to deliver a biological effect and desired phenotype in model systems. Despite the challenges outlined by the authors, other modalities described (oligonucleotide, small molecule) have been proven in this context. Furthermore, significant work has already been done to describe the biological effect seen when using bioPROTACs against RAS, just not when delivered as a protein (work by the Rabbits and Koide groups), and as such provide a valuable benchmark.

We have included an additional study on LNP:bioPROTACs in native MIA PaCa-2 PDAC cell lines to explore Ras degradation in a disease-relevant cell line. This degradation results in a simultaneous decrease in Erk phosphorylation consistent with MAPK pathway inhibition. Finally, Ras degradation leads to anti-proliferative effects in this cell line. See newly-added Figures: 9, S14 and S15. Like previous work by the Rabbits and Koide group, our findings confirm that Ras degradation leads inhibition of Ras-mutated cells. Unlike previous studies, however, we demonstrate this effect by protein delivery rather than by nucleic acid methods or bioPROTAC transgene induction.

A more specific area where the manuscript could be improved is the inclusion of orthogonal methods to validate and support the authors observations. For example, global proteomics has been applied to investigate PROTAC on- and off-target effects. The paper would benefit from such an approach, to link the loss of the target protein to a pathway effect in disease relevant cell lines, and any perturbations that might result from using a protein-based delivery system versus nucleic acid, for example. Kinetic studies could also be undertaken.

School of Engineering and Applied Science
Department of Bioengineering
240 Skirkanich Hall
210 South 33rd Street
Philadelphia, PA 19104-6321
Tel 215.898.8501 Fax 215.573.3155

We thank the reviewer for this suggestion, and we have performed global proteomic profiling perturbations specific to mRNA or protein alone. Data are included as Figure 7, Supplemental Figure 11, and further discussion. Firstly, we found that Ras downregulation by LC-MS/MS agrees with western blot analysis. However, mRNA displays many additional knockdowns compared to protein treatment (50% of all knockdowns are unique to mRNA). Further investigation found that some of these proteins were likely cellular responses to foreign RNA. However, we identified 12 GTPases uniquely downregulated by mRNA bioPROTAC. In-depth comparison of these GTPases suggests that some are also possible targets of protein bioPROTAC due to correlated log fold changes. However, we identified alpha and beta tubulin subunits as uniquely downregulated by mRNA with no correlation in the corresponding protein delivery group, suggesting these could be modality-specific off targets. Moreover, we identified splice factors as downregulated in null bioPROTAC and mRNA bioPROTAC groups. We attribute this to the lpaH9.8 domain which has been shown to bind to and inhibit U2AF splice factors. Additional discussion regarding this finding is included in the main text.

Also, it wasn't clear from the text if the authors had removed native substrate binding capacity for relevant E3s, as this could have a marked effect on the specificity of these construct.

This is an important clarification that should be addressed. Indeed we removed the native substrate domain of all E3 domains tested. We further clarified this point in the main text to avoid confusion:

For all E3 proteins tested, only the degradation domains capable of recruiting E3 complex proteins or E2 conjugating enzymes were used, and the native substrate binding portions were

not included in bioPROTAC designs.

In terms of target turnover and mechanism, native cell lines should also be used to confirm / explore kinetics and hook effects.

We performed additional testing in native cell lines. Specifically, we performed dose response studies in a PDAC model, MIA-PaCa and found no hook effect for protein or mRNA bioPROTAC (see Figure 9). In fact, the only time we have observed a hook effect is with plasmid transfection. It is likely that our optimized pcDNA plasmids including a Kozak sequence and a strong CMV promoter produced excessive bioPROTAC production in HEK293T cells leading to the initial observed hook effect. Other modalities likely cannot reach the same protein levels, and apparent hook effects were not observed in any other context.

School of Engineering and Applied Science
Department of Bioengineering
240 Skirkanich Hall
210 South 33rd Street
Philadelphia, PA 19104-6321
Tel 215.898.8501 Fax 215.573.3155

Furthermore, the mechanism of action was confirmed by target ubiquitylation using a biochemical assay with purified proteins, but no data was shown in the cellular context, either by inhibiting the ability for the target to be processed through the proteasomal pathway once ubiquitylated, or removal of the bioPROTAC core activity by mutation to prevent ubiquitin addition.

We thank the reviewer for this helpful addition. We conducted an additional mechanistic study by either mutating the catalytic cysteine of IpaH9.8 and/or adding the proteasome inhibitor, MG-132, to cell culture media (see supplemental figures 12 and 13). We found that abolishing bioPROTAC catalytic activity completely rescued target degradation of both overexpressed GFP-KRAS and endogenous Ras. MG-132 completely rescued endogenous Ras and partially rescued GFP-KRAS. We conclude that our bioPROTACs do proceed through the UPS mechanism. Additional pathways such as endosome-lysosome or autophagy might also partake in degradation, but if they do, they are ubiquitin-dependent mechanisms.

Also, turnover of the bioPROTAC molecule was not investigated - is the bioPROTAC degraded with the target or can it act catalytically? Is the construct turned over without target engagement? With the current format, if the bioPROTAC acts stoichiometrically this could limit the ability to generate and deliver sufficient material to provide a sustained biological effect.

The reviewer asks an important question regarding the fate of our bioPROTACs once delivered into cells. First, we note that the catalytic domain of IpaH9.8 is known to auto-ubiquitinate and contains an auto-inhibitory domain in the LRR domain of the full-length protein (ref 63). This auto-inhibitory domain is likely important to prevent premature degradation of the E3 ligase during host cell immune modulation.

We performed further experiments and show that our bioPROTACs auto-ubiquitinate (Figure S12). In addition, fluorescent readout by split GFP is markedly lower in GFP(1-10) reporter cells treated with wild-type IpaH9.8 compared to the C337A variant (catalytically dead) indicating that the protein's catalytic activity leads to its destruction in the cytosol (Figure S13). We suspect that the majority of the auto-degradation is independent of substrate binding, as we observed no difference in the split GFP signal between IpaHWT-K27 and IpaHWT-K27n3. If substrate binding were critical to self-destruction, we might expect that the IpaHWT-K27n3 split GFP signal would be higher, as it is unable to bind to endogenous Ras. Finally, as MG-132 did not rescue GFP complementation, it is possible that auto-ubiquitination leads to degradation through a non-proteasomal pathway such as autophagy.

School of Engineering and Applied Science
Department of Bioengineering
240 Skirkanich Hall
210 South 33rd Street
Philadelphia, PA 19104-6321
Tel 215.898.8501 Fax 215.573.3155

Ultimately, our bioPROTACs are subject to autoregulation and their therapeutic potencies may be affected by this auto-ubiquitination. Despite this observed self-degradation, we

still report rapid and potent degradation of targets in vitro. In future studies, we aim to stabilize this molecule for in vivo applications by identifying and selectively mutating self-ubiquitinated lysine residues, rendering bioPROTACs resistant or immune to autoubiquitination.

Furthermore, our data indicate that the bioPROTACs are behaving catalytically rather than stoichiometrically. Firstly, delivery of the catalytically-dead lpaH9.8C337A-K27-D25-s11 (100nM) in GFP(1-10) cells produced a fluorescent signal with 58% normalized mean intensity relative to untreated EGFP-KRAS overexpression cells (shown below). This is despite inducing ~95% target degradation. Hence, the protein amount delivered is less than the degradation efficiency, pointing to catalytic activity.

We also note that splitGFP is derived from superfolder GFP which is markedly brighter than EGFP (54,000 versus 34,000) making the true difference even larger than what we are noting here. Reconstituted splitGFP has been shown to be as bright as full length superfolder GFP (Feng, S., Sekine, S., Pessino, V. et al. Improved split fluorescent proteins for endogenous protein labeling. Nat Commun 8, 370 (2017). This allows us to make a reasonable comparison between EGFP (target) and splitGFP (delivery reporter). Further, we are not saturating the pool of GFP(1-10) in 293T reporter cells, as we previously showed even higher complementation levels in the same 293T GFP(1-10) cells with another s11-tagged protein at shorter treatment time lengths than described here (ref 40)

In a similar line of reasoning, the total concentration of endogenous Ras in 293T cells has been estimated to be approximately 300nM (ref 77). Our GFP-KRAS construct is overexpressed well above this endogenous level, and treatment with only 100nM of bioPROTAC can almost completely clear the target.

School of Engineering and Applied Science
Department of Bioengineering
240 Skirkanich Hall
210 South 33rd Street
Philadelphia, PA 19104-6321
Tel 215.898.8501 Fax 215.573.3155

[Redacted]

Taken together, these data suggest that our bioPROTACs are catalytic, as efficient degradation is achieved at sub-stoichiometric levels.

Regarding the presentation of the manuscript itself, there is a lack of labelling of figures and provision of figure legends. Also, the reporting summary is brief and indicates cells lines were not validated, which is of concern. These elements need more attention.

Figure captions and labels have been added, and all native cell lines used in this study and those used to generate reporter cell lines have been authenticated by STR profiling.

Overall, the paper is very interesting but requires additional data to validate the approach, as outlined above. Crucially, further exploration of the biological effect observed when using these tools in relevant pre-clinical models is required—how well do they perform in this context? I expect this will be how most readers/users would want to apply this technology, so a more detailed exploration and validation is essential for maximum impact, ideally with it benchmarked against other delivery systems. Proof-of-concept for in vivo delivery would be transformative.

School of Engineering and Applied Science
Department of Bioengineering
240 Skirkanich Hall
210 South 33rd Street
Philadelphia, PA 19104-6321
Tel 215.898.8501 Fax 215.573.3155

The authors speculate on future therapeutic potential, but much work needs to be done to understand the efficacy of the constructs described, in addition to their tolerability, before this could even be considered.

We have included three additional studies exploring proteomic changes, inhibitory effects in an in vitro PDAC model, and degradation mechanisms to explore the biological effects of protein bioPROTACs. Moreover, we benchmarked these experiments against bioPROTAC mRNA delivery, a common modality for the field. We believe animal experiments are outside the scope of this study. Additional methods development including targeted protein LNPs and

bioPROTAC stabilization might be necessary before undertaking in vivo work and are better suited for a standalone study.

Version 1:

Reviewer comments:

Reviewer #1

(Remarks to the Author)

In this revised manuscript, the authors provided additional description and information to clarify the experiment procedures and data analysis. New experimental data support the experimental design and conclusions.

Reviewer #3

(Remarks to the Author)

For **Responses** and further *reviewer* comments please see below.

Reviewer #3: Chan et al describe a system for the intracellular delivery of bioPROTACs as recombinant proteins, concentrating on targeting RAS. To-date approaches to deliver this type of molecule have focused on mRNA, plasmid DNA and viral methods. The ability to deliver bioPROTACs without the requirement for genetic encoding opens up experimental and therapeutic opportunities for their application. As such, it is an exciting and worthwhile area of research.

The work described in the manuscript does not provide a significant addition to the current bioPROTAC literature in terms of targets or design, applying elements of previously described tools. The observation that an lpaH9.8 bioPROTAC is the most effective in the context of this approach is interesting, but not investigated in detail. At the end of the paper the target space is expanded beyond RAS, which is tantalising, but this part of the paper is underdeveloped. The authors indicate that they have described the delivery technology before for other biologic formats, and therefore this is also not novel outside of the bioPROTAC context.

Although promising, the data presented by the authors is somewhat preliminary, focusing for the most-part on the depletion of over-expressed, as opposed to native protein. The effect on endogenous RAS levels in disease-relevant cancer cell lines was much less pronounced and confined to the supplementary data section.

Given the lack of novelty for the constituent elements of the bioPROTAC system described, the study should be expanded into a more thorough analysis of the opportunity that this delivery approach affords with respect to modulating the activity of native proteins classed as “undruggable”, specifically the ability to deliver a biological effect and desired phenotype in model systems. Despite the challenges outlined by the authors, other modalities described (oligonucleotide, small molecule) have been proven in this context. Furthermore, significant work has already been done to describe the biological effect seen when using bioPROTACs against RAS, just not when delivered as a protein (work by the Rabbitts and Koide groups), and as such provide a valuable benchmark.

We have included an additional study on LNP:bioPROTACs in native MIA PaCa-2 PDAC cell lines to explore Ras degradation in a disease-relevant cell line. This degradation results in a simultaneous decrease in Erk phosphorylation consistent with MAPK pathway inhibition. Finally, Ras degradation leads to anti-proliferative effects in this cell line. See newly-added Figures: 9, S14 and S15. Like previous work by the Rabbitts and Koide group, our findings confirm that Ras degradation leads inhibition of Ras-mutated cells. Unlike previous studies, however, we demonstrate this effect by protein delivery rather than by nucleic acid methods or bioPROTAC transgene induction.

The efforts made by the authors to address this comment are appreciated and it is encouraging to see that delivery of the bioPROTAC as a protein construct has an anti-proliferative effect. However, it does raise further questions as to the mechanism. The null variant (K27n3) also appears to show a dose-dependent effect (Fig. 9D), with at least half of the bioPROTAC maximal effect not the result of depleting RAS but possibly due to other properties, such as those of the delivery system, E3 ligase alone and/or the inhibitory effect of the DARPIN. In the main manuscript supplementary data is highlighted that shows changing formulation does alter the profile, but these data are not analysed for statistical significance. The supplementary and main figures appear to be a combination of technical and biological repeats, these should be presented / analysed separately and these experiments would also benefit from additional controls, for example using a bioPROTAC with an inactive E3 and a dual knockout (E3 catalytically inactive and K27n3). Also, the fact that reformulation is required to manage cytotoxic effects adds a degree of complexity if wanting to apply the technique generally. Furthermore, mechanistically it is important to show that the bioPROTAC construct inhibits proliferation by inducing programmed cell death, which could be achieved by analysing samples for apoptosis markers, for example.

A more specific area where the manuscript could be improved is the inclusion of orthogonal methods to validate and support the authors observations. For example, global proteomics has been applied to investigate PROTAC on- and off-target effects. The paper would benefit from such an approach, to link the loss of the target protein to a pathway effect in disease relevant cell lines, and any perturbations that might result from using a protein-based delivery system versus nucleic acid, for example. Kinetic studies could also be undertaken.

We thank the reviewer for this suggestion, and we have performed global proteomic profiling perturbations specific to mRNA or protein alone. Data are included as Figure 7, Supplemental Figure 11, and further discussion. Firstly, we found that Ras downregulation by LC-MS/MS agrees with western blot analysis. However, mRNA displays many additional knockdowns compared to protein treatment (50% of all knockdowns are unique to mRNA). Further investigation found that some of these proteins were likely cellular responses to foreign RNA. However, we identified 12 GTPases uniquely downregulated by mRNA bioPROTAC. In-depth comparison of these GTPases suggests that some are also possible targets of protein bioPROTAC due to correlated log fold changes. However, we identified alpha and beta tubulin subunits as uniquely downregulated by mRNA with no correlation in the corresponding protein delivery group, suggesting these could be modality-specific off targets. Moreover, we identified splice factors as downregulated in null bioPROTAC and mRNA bioPROTAC groups. We attribute this to the IpaH9.8 domain which has been shown to bind to and inhibit U2AF splice factors. Additional discussion regarding this finding is included in the main text.

The authors efforts to address this question are appreciated. However, my previous questions is only partially addressed as the HEK cell line was chosen. The disease relevant cell line used in the proliferation experiments would represent a better model to deliver an understanding of the pathway specific effects, this would have also helped to provide data supportive to the observation made in the proliferation experiments that inhibition of the RAS pathway by TPD was the cause of the anti-proliferative effect. See previous comments. Also, the null (non-binding) construct appears to have a very significant effect on the proteome, again highlighting potential off-target effects of the construct; the level of RAS depletion is modest in comparison to some of the other proteome changes observed.

Also, it wasn't clear from the text if the authors had removed native substrate binding capacity for relevant E3s, as this could have a marked effect on the specificity of these construct.

This is an important clarification that should be addressed. Indeed we removed the native substrate domain of all E3 domains tested. We further clarified this point in the main text to avoid confusion: "For all E3 proteins tested, only the degradation domains capable of recruiting E3 complex proteins or E2 conjugating enzymes were used, and the native substrate binding portions were not included in bioPROTAC designs."

Thank you for clarifying.

In terms of target turnover and mechanism, native cell lines should also be used to confirm / explore kinetics and hook effects.

We performed additional testing in native cell lines. Specifically, we performed dose response studies in a PDAC model, MIA-PaCa and found no hook effect for protein or mRNA bioPROTAC (see Figure 9). In fact, the only time we have observed a hook effect is with plasmid transfection. It is likely that our optimized pcDNA plasmids including a Kozak sequence and a strong CMV promoter produced excessive bioPROTAC production in HEK293T cells leading to the initial observed hook effect. Other modalities likely cannot reach the same protein levels, and apparent hook effects were not observed in any other context.

Thank you for clarifying.

Furthermore, the mechanism of action was confirmed by target ubiquitylation using a biochemical assay with purified proteins, but no data was shown in the cellular context, either by inhibiting the ability for the target to be processed through the proteasomal pathway once ubiquitylated, or removal of the bioPROTAC core activity by mutation to prevent ubiquitin addition.

We thank the reviewer for this helpful addition. We conducted an additional mechanistic study by either mutating the catalytic cysteine of IpaH9.8 and/or adding the proteasome inhibitor, MG-132, to cell culture media (see supplemental figures 12 and 13). We found that abolishing bioPROTAC catalytic activity completely rescued target degradation of both overexpressed GFP-KRAS and endogenous Ras. MG-132 completely rescued endogenous Ras and partially rescued GFP-KRAS. We conclude that our bioPROTACs do proceed through the UPS mechanism. Additional pathways such as endosome-lysosome or autophagy might also partake in degradation, but if they do, they are ubiquitin-dependent mechanisms.

Thank you to the authors for making efforts to address this question. Did the authors consider using lysosomal inhibitors to better understand the contribution of lysosomal processes to the mechanism of action? The proteomic study appears to indicate lysosomal degradation to be important. All these observations suggest a complex system at play with this molecule, with elements not related to classic UPS pathway-mediated processes and possible significant contributions from off-pathway effects.

Also, turnover of the bioPROTAC molecule was not investigated – is the bioPROTAC degraded with the target or can it act catalytically? Is the construct turned over without target engagement? With the current format, if the bioPROTAC acts stoichiometrically this could limit the ability to generate and deliver sufficient material to provide a sustained biological effect.

The reviewer asks an important question regarding the fate of our bioPROTACs once delivered into cells. First, we note that the catalytic domain of IpaH9.8 is known to autoubiquitinate and contains an auto-inhibitory domain in the LRR domain of the full-length protein (ref 63). This auto-inhibitory domain is likely important to prevent premature degradation of the E3

ligase during host cell immune modulation. We performed further experiments and show that our bioPROTACs auto-ubiquitinate (Figure S12). In addition, fluorescent readout by split GFP is markedly lower in GFP(1-10) reporter cells treated with wild-type IpaH9.8 compared to the C337A variant (catalytically dead) indicating that the protein's catalytic activity leads to its destruction in the cytosol (Figure S13). We suspect that the majority of the auto-degradation is independent of substrate binding, as we observed no difference in the split GFP signal between IpaHWT-K27 and IpaHWT-K27n3. If substrate binding were critical to self-destruction, we might expect that the IpaHWT-K27n3 split GFP signal would be higher, as it is unable to bind to endogenous Ras. Finally, as MG-132 did not rescue GFP complementation, it is possible that autoubiquitination leads to degradation through a non-proteasomal pathway such as autophagy.

Ultimately, our bioPROTACs are subject to autoregulation and their therapeutic potencies may be affected by this autoubiquitination. Despite this observed self-degradation, we still report rapid and potent degradation of targets in vitro. In future studies, we aim to stabilize this molecule for in vivo applications by identifying and selectively mutating self-ubiquitinated lysine residues, rendering bioPROTACs resistant or immune to autoubiquitination. Furthermore, our data indicate that the bioPROTACs are behaving catalytically rather than stoichiometrically. Firstly, delivery of the catalytically-dead IpaH9.8C337A-K27-D25-s11 (100nM) in GFP(1-10) cells produced a fluorescent signal with 58% normalized mean intensity relative to untreated EGFP-KRAS overexpression cells (shown below). This is despite inducing ~95% target degradation. Hence, the protein amount delivered is less than the degradation efficiency, pointing to catalytic activity. We also note that splitGFP is derived from superfolder GFP which is markedly brighter than EGFP (54,000 versus 34,000) making the true difference even larger than what we are noting here. Reconstituted splitGFP has been shown to be as bright as full length superfolder GFP (Feng, S., Sekine, S., Pessino, V. et al. Improved split fluorescent proteins for endogenous protein labeling. Nat Commun 8, 370 (2017). This allows us to make a reasonable comparison between EGFP (target) and splitGFP (delivery reporter). Further, we are not saturating the pool of GFP(1-10) in 293T reporter cells, as we previously showed even higher complementation levels in the same 293T GFP(1-10) cells with another s11-tagged protein at shorter treatment time lengths than described here (ref 40) In a similar line of reasoning, the total concentration of endogenous Ras in 293T cells has been estimated to be approximately 300nM (ref 77). Our GFP-KRAS construct is overexpressed well above this endogenous level, and treatment with only 100nM of bioPROTAC can almost completely clear the target. Taken together, these data suggest that our bioPROTACs are catalytic, as efficient degradation is achieved at sub-stoichiometric levels.

The authors highlight important observations, but the evidence relies on measuring relative levels of fluorescence which could be influenced by a range of factors. Further studies will be required to definitively demonstrate a catalytic mechanism of action; auto-degradation argues against this MOA, experiments to minimise this process would help to support claims of a catalytic process but additional efforts are needed, for example absolute measurements of bioPROTAC and substrate levels. Furthermore, the fact that autophagy may play role again indicates the complexity of the mechanism associated with this construct and that off-pathway perturbations could be significant interfering factors that limit the wider application of the system described.

Regarding the presentation of the manuscript itself, there is a lack of labelling of figures and provision of figure legends. Also, the reporting summary is brief and indicates cells lines were not validated, which is of concern. These elements need more attention.

Figure captions and labels have been added, and all native cell lines used in this study and those used to generate reporter cell lines have been authenticated by STR profiling.

Thank you for confirming. In addition, further efforts are required to add information regarding sample sizes to figure legends, this is not clear in all cases, and to make clear how these data are treated in terms of their statistical comparison in the reporting summary, for example whether they represent technical or biological repeats.

Overall, the paper is very interesting but requires additional data to validate the approach, as outlined above. Crucially, further exploration of the biological effect observed when using these tools in relevant pre-clinical models is required– how well do they perform in this context? I expect this will be how most readers/users would want to apply this technology, so a more detailed exploration and validation is essential for maximum impact, ideally with it benchmarked against other delivery systems. Proof-of-concept for in vivo delivery would be transformative.

The authors speculate on future therapeutic potential, but much work needs to be done to understand the efficacy of the constructs described, in addition to their tolerability, before this could even be considered.

We have included three additional studies exploring proteomic changes, inhibitory effects in an in vitro PDAC model, and degradation mechanisms to explore the biological effects of protein bioPROTACs. Moreover, we benchmarked these experiments against bioPROTAC mRNA delivery, a common modality for the field. We believe animal experiments are outside the scope of this study. Additional methods development including targeted protein LNPs and bioPROTAC stabilization might be necessary before undertaking in vivo work and are better suited for a standalone study.

Thank you to the authors for carrying out these additional experiments and I agree that in vivo studies are outside of the scope of this study. The new data presented adds to the manuscript and it is exciting to see a protein-based system for the delivery of bioPROTACs. In my view, the manuscript is comprehensive but there remain questions regarding the mechanism of action, particularly how much off-target processes contribute to the anti-proliferative properties observed and the role played by autoubiquitylation and lysosomal degradation. Until the impact of these is better understood it could limit the application of the platform as a tool for proteome editing.

May 2, 2024

We thank the reviewers for the thoughtful comments and suggestions to strengthen our manuscript. We have edited our manuscript accordingly and have addressed each point below.

Reviewer #1:

Comments/suggestions following submission of our revised manuscript: In this revised manuscript, the authors provided additional description and information to clarify the experiment procedures and data analysis. New experimental data support the experimental design and conclusions.

We thank the reviewer for their comment and are pleased that the reviewer finds their comments from their initial review adequately addressed.

Reviewer #2:

Comments/suggestions following submission of our revised manuscript: None

We are pleased that we were able to satisfactorily address all of the reviewers comments.

Reviewer #3 (Since many of the reviewer comments/suggestions are related to their initial review and our response, we have included these entire discussions in our response. Initial comments are in plain text. Our response to the reviewer's initial comments is in bold. The reviewer's comments that followed submission of our revised manuscript are in red. Our response to these comments is highlighted.):

1. Comments/suggestions following submission of our initial submission: Chan et al describe a system for the intracellular delivery of bioPROTACs as recombinant proteins, concentrating on targeting RAS. To-date approaches to deliver this type of molecule have
School of Engineering and Applied Science
Department of Bioengineering
240 Skirkanich Hall
210 South 33rd Street
Philadelphia, PA 19104-6321
Tel 215.898.8501 Fax 215.573.3155

focused on mRNA, plasmid DNA and viral methods. The ability to deliver bioPROTACs without the requirement for genetic encoding opens up experimental and therapeutic opportunities for their application. As such, it is an exciting and worthwhile area of research.

The work described in the manuscript does not provide a significant addition to the current bioPROTAC literature in terms of targets or design, applying elements of previously described tools. The observation that an lpaH9.8 bioPROTAC is the most effective in the context of this approach is interesting, but not investigated in detail. At the end of the paper the target space is expanded beyond RAS, which is tantalising, but this part of the paper is underdeveloped. The authors indicate that they have described the delivery technology

before for other biologic formats, and therefore this is also not novel outside of the bioPROTAC context.

Although promising, the data presented by the authors is somewhat preliminary, focusing for the most-part on the depletion of over-expressed, as opposed to native protein. The effect on endogenous RAS levels in disease-relevant cancer cell lines was much less pronounced and confined to the supplementary data section.

Given the lack of novelty for the constituent elements of the bioPROTAC system described, the study should be expanded into a more thorough analysis of the opportunity that this delivery approach affords with respect to modulating the activity of native proteins classed as "undruggable", specifically the ability to deliver a biological effect and desired phenotype in model systems. Despite the challenges outlined by the authors, other modalities described (oligonucleotide, small molecule) have been proven in this context. Furthermore, significant work has already been done to describe the biological effect seen when using bioPROTACs against RAS, just not when delivered as a protein (work by the Rabbits and Koide groups), and as such provide a valuable benchmark.

Our initial response to the reviewer's comments: We have included an additional study on LNP:bioPROTACs in native MIA PaCa-2 PDAC cell lines to explore Ras degradation in a disease-relevant cell line. This degradation results in a simultaneous decrease in Erk phosphorylation consistent with MAPK pathway inhibition. Finally, Ras degradation leads to anti-proliferative effects in this cell line. See newly-added Figures: 9, S14 and S15. Like previous work by the Rabbits and Koide group, our findings confirm that Ras degradation leads to inhibition of Ras-mutated cells. Unlike previous studies, however, we demonstrate this effect by protein delivery rather than by nucleic acid methods or bioPROTAC transgene induction.

Comments/suggestions following submission of our revised manuscript: The efforts made by the authors to address this comment are appreciated and it is encouraging to see that delivery of the bioPROTAC as a protein construct has an anti-proliferative effect. However, it does raise further questions as to the mechanism. The null variant (K27n3) also appears to show a dose-dependent effect (Fig. 9D), with at least half of the bioPROTAC maximal effect

School of Engineering and Applied Science
Department of Bioengineering

240 Skirkanich Hall

210 South 33rd Street

Philadelphia, PA 19104-6321

Tel 215.898.8501 Fax 215.573.3155

not the result of depleting RAS but possibly due to other properties, such as those of the delivery system, E3 ligase alone and/or the inhibitory effect of the DARPIN. In the main manuscript supplementary data is highlighted that shows changing formulation does alter the profile, but these data are not analysed for statistical significance. The supplementary and main figures appear to be a combination of technical and biological repeats, these should be presented / analysed separately and these experiments would also benefit from additional controls, for example using a bioPROTAC with an inactive E3 and a dual knockout (E3 catalytically inactive and K27n3). Also, the fact that reformulation is required to manage cytotoxic effects adds a degree of complexity if wanting to apply the technique generally. Furthermore, mechanistically it is important to show that the bioPROTAC construct inhibits proliferation by inducing programmed cell death, which could be achieved by analysing samples for apoptosis markers, for example.

Response to latest comments/suggestions: We have reformatted the data to only contain biological replicates and have performed statistical tests for all RTCA experiments including Figure S15. We have repeated the MIA PaCa-2 proliferation experiment an additional 4 times. We have included two additional protein controls to fully interrogate the system. Firstly, we added a treatment group receiving the C337A catalytically-dead mutant fused to the Ras-binding K27 DARPIn. Secondly, we cloned, expressed, and purified a double-negative control containing both an inactive IpaH9.8 domain and the non-binding DARPIn K27n3. All proteins were formulated in the K1 LNP formulation. The results of the proliferation assay have been added to main Figure 9 and supporting modifications have been made to S14 and S15. Of all the tested proteins, only the active bioPROTAC was able to degrade Ras and inhibit MAPK signaling as detected by western blotting. Interestingly, IpaH9.8(C337A)-K27 was not effective in blocking Erk phosphorylation despite its demonstrated ability to bind Ras (Supplemental Figure S12). This is likely due to inadequate delivery relative to MIA PaCa-2 Ras expression levels. We chose to compare all proteins at a low dose to avoid confounding off-target toxicities and found that the active bioPROTAC treatment group significantly reduced MIA PaCa-2 proliferation. The combination of the 4 protein controls and RTCA experiments provides evidence that Ras degradation is indeed responsible for PDAC cell growth inhibition. As suggested by the reviewer, we used two antibodies to probe for cleaved caspase 3 following LNP treatment but did not detect elevated levels of the apoptotic marker (data not shown). This indicates that decreased cell

proliferation was simply a result of slowed growth relative to control groups. A previous report had demonstrated increased cleaved caspase 3 (Ref 26) using an integrated Tet-on inducible bioPROTAC system, but this approach was able to completely degrade Ras (nearly 100% efficiency) leading to near-complete MAPK inhibition. In contrast, we were able to degrade Ras by ~50-60% with a comparable level of MAPK inhibition. This difference in Ras degradation could explain the differing biological outcomes observed as well as our modest growth inhibition results. Other studies using small-molecules and siRNA have also found that apoptosis markers are not detected at low-to-moderate MAPK

School of Engineering and Applied Science

Department of Bioengineering

240 Skirkanich Hall

210 South 33rd Street

Philadelphia, PA 19104-6321

Tel 215.898.8501 Fax 215.573.3155

inhibition levels^{1,2}. Nonetheless, we consistently observed decreased proliferation in PDAC cells treated with the active bioPROTAC compared to other protein controls. Finally, we agree with the reviewer's observations regarding reformulation and potential off-target toxicities, as this seems to be the major source of non-specific inhibition. Future in vivo studies should focus on simultaneously designing high-efficiency bioPROTACs and minimally-toxic nanocarriers for maximal anti-cancer effects, and we have already stated this in the main text.

2. Comments/suggestions following submission of our initial submission: A more specific area where the manuscript could be improved is the inclusion of orthogonal methods to validate and support the authors observations. For example, global proteomics has been applied to investigate PROTAC on- and off-target effects. The paper would benefit from such an approach, to link the loss of the target protein to a pathway effect in disease relevant cell lines, and any perturbations that might result from using a protein-based delivery system versus nucleic acid, for example. Kinetic studies could also be undertaken.

Our initial response to the reviewer's comments: We thank the reviewer for this suggestion, and we have performed global proteomic profiling perturbations specific to mRNA or protein alone. Data are included as Figure 7, Supplemental Figure 11, and further discussion. Firstly, we found that Ras downregulation by LC-MS/MS agrees with western blot analysis. However, mRNA displays many additional knockdowns compared to protein treatment (50% of all knockdowns are unique to mRNA). Further investigation found that some of these proteins were likely cellular responses to foreign RNA. However, we identified 12 GTPases uniquely downregulated by mRNA bioPROTAC. In-depth comparison of these GTPases suggests that some are also possible targets of protein bioPROTAC due to correlated log fold changes. However, we identified alpha and beta tubulin subunits as uniquely downregulated by mRNA with no correlation in the corresponding protein delivery group, suggesting these could be modality-specific off targets. Moreover, we identified splice factors as downregulated in null bioPROTAC and mRNA bioPROTAC groups. We attribute this to the IpaH9.8 domain which has been shown to bind to and inhibit U2AF splice factors. Additional discussion regarding this finding is included in the main text.

Comments/suggestions following submission of our revised manuscript: The authors efforts to address this question are appreciated. However, my previous questions is only partially addressed as the HEK cell line was chosen. The disease relevant cell line used in the proliferation experiments would represent a better model to deliver an understanding of the pathway specific effects, this would have also helped to provide data supportive to the observation made in the proliferation experiments that inhibition of the RAS pathway by TPD was the cause of the anti-proliferative effect. See previous comments. Also, the null (non-

School of Engineering and Applied Science

Department of Bioengineering

240 Skirkanich Hall

210 South 33rd Street

Philadelphia, PA 19104-6321

Tel 215.898.8501 Fax 215.573.3155

binding) construct appears to have a very significant effect on the proteome, again highlighting potential off-target effects of the construct; the level of RAS depletion is modest in comparison to some of the other proteome changes observed.

Response to latest comments/suggestions: We understand the intention of the

reviewer's suggestion to link changes in complex signaling pathways to a biological phenotype (decreased proliferation). However, after careful consideration, we have decided not to carry out this experiment. Firstly, pathway perturbations would be cell-type specific, even between Ras-dependent cancer cells. Thus, analysis in MIA PaCA-2 would not be generally useful to other cell lines regarding cell proliferation. Here, the reviewer is proposing a much more in-depth pathway analysis that would require extensive sample preparation optimization (for LC-MS). Conclusions drawn from these efforts would only apply to a very narrow application of the technology we are describing. Moreover, to fully satisfy the reviewer's curiosity, supporting experiments such as transcriptomics (with their own optimizations procedures) would likely need to be performed. In light of this, it seems that this particular recommendation is in the realm of a separate manuscript. The complexity of Ras, a master regulator, would all but guarantee that this is not just one experiment, as the reviewer is implying, but rather a host of studies to fully probe Ras-related pathways following bioPROTAC treatment^{3,4}. Moreover, signaling cascades, being dynamic processes would require us to perform time-course proteomics/transcriptomics studies. Such an extension would not be suitable for the length and context of this manuscript, and for these reasons, we have decided to exclude such studies.

Regarding anti-proliferative effects. We have now included 4 proteins in RTCA experiments, mutating either the E3 or DARPIn domains individually or both simultaneously as suggested by the reviewer. In addition, we have measured the dose-dependent effects of each of these proteins on both Ras and pERK levels (to probe canonical MAPK signaling). We have now shown that both Ras degradation and decreased pERK are linked to reduced proliferation. The MIA PaCa-2 cell line is widely accepted to be a Ras-dependent. Thus, we strongly believe our data sufficiently supports the mechanism of growth inhibition without the need for an exhaustive proteomics experiment. As the reviewer points out, the null bioPROTAC did have some unexpected effects, although they are not necessarily off-target. As discussed in the prior response, supplemental data, as well as in the main text, the IpaH9.8 NEL is known to bind U2AF. We recapitulated this effect via proteomics and have already provided relevant discussion for future users to take into consideration. We note that appending a target-specific DARPIn (K27) reduced splice factor downregulation, likely due to redirection of the bioPROTAC.

In the initial comment, the reviewer had provided an insightful suggestion to compare the effects between bioPROTAC modalities. We found this helpful and elected on using HEK293 to compare the effects of different bioPROTAC modalities. We believe this to be an

School of Engineering and Applied Science
Department of Bioengineering
240 Skirkanich Hall
210 South 33rd Street
Philadelphia, PA 19104-6321
Tel 215.898.8501 Fax 215.573.3155

appropriate cell line and most salient application of proteomics to validate our system. Firstly, HEK293T are among the most commonly-used cells in research labs. Secondly, we used HEK293T throughout the entire manuscript, so conclusions drawn from the proteomics experiment can be tied to other aspects of the bioPROTAC technology. We believe that our current proteomic study better validates the platform technology introduced here.

3. Comments/suggestions following submission of our initial submission: Also, it wasn't clear from the text if the authors had removed native substrate binding capacity for relevant E3s, as this could have a marked effect on the specificity of these construct.

Our initial response to the reviewer's comments: This is an important clarification that should be addressed. Indeed we removed the native substrate domain of all E3 domains tested. We further clarified this point in the main text to avoid confusion: "For all E3 proteins tested, only the degradation domains capable of recruiting E3 complex proteins or E2 conjugating enzymes were used, and the native substrate binding portions were not included in bioPROTAC designs."

Comments/suggestions following submission of our revised manuscript: Thank you for clarifying.

Response to latest comments/suggestions: We are pleased that the reviewer finds this question was adequately addressed.

4. Comments/suggestions following submission of our initial submission: In terms of target turnover and mechanism, native cell lines should also be used to confirm / explore kinetics and hook effects.

Our initial response to the reviewer's comments: We performed additional testing in native cell lines. Specifically, we performed dose response studies in a PDAC model, MIA-PaCa and found no hook effect for protein or mRNA bioPROTAC (see Figure 9).

In fact, the only time we have observed a hook effect is with plasmid transfection. It is likely that our optimized pcDNA plasmids including a Kozak sequence and a strong CMV promoter produced excessive bioPROTAC production in HEK293T cells leading to the initial observed hook effect. Other modalities likely cannot reach the same protein levels, and apparent hook effects were not observed in any other context. Comments/suggestions following submission of our revised manuscript: Thank you for clarifying.

Response to latest comments/suggestions: We are pleased that the reviewer finds this question was adequately addressed.

School of Engineering and Applied Science
Department of Bioengineering
240 Skirkanich Hall
210 South 33rd Street
Philadelphia, PA 19104-6321
Tel 215.898.8501 Fax 215.573.3155

5. Comments/suggestions following submission of our initial submission: Furthermore, the mechanism of action was confirmed by target ubiquitylation using a biochemical assay with purified proteins, but no data was shown in the cellular context, either by inhibiting the ability for the target to be processed through the proteasomal pathway once ubiquitylated, or removal of the bioPROTAC core activity by mutation to prevent ubiquitin addition.

Our initial response to the reviewer's comments: We thank the reviewer for this helpful addition. We conducted an additional mechanistic study by either mutating the catalytic cysteine of IpaH9.8 and/or adding the proteasome inhibitor, MG-132, to cell culture media (see supplemental figures 12 and 13). We found that abolishing bioPROTAC catalytic activity completely rescued target degradation of both overexpressed GFP-KRAS and endogenous Ras. MG-132 completely rescued endogenous Ras and partially rescued GFP-KRAS. We conclude that our bioPROTACs do proceed through the UPS mechanism. Additional pathways such as endosome-lysosome or autophagy might also partake in degradation, but if they do, they are ubiquitin-dependent mechanisms.

Comments/suggestions following submission of our revised manuscript: Thank you to the authors for making efforts to address this question. Did the authors consider using lysosomal inhibitors to better understand the contribution of lysosomal processes to the mechanism of action? The proteomic study appears to indicate lysosomal degradation to be important. All these observations suggest a complex system at play with this molecule, with elements not related to classic UPS pathway-mediated processes and possible significant contributions from off-pathway effects.

Response to latest comments/suggestions: Initially, we did not anticipate that other mechanisms beyond the UPS would be involved in target degradation. However, to further shed light on whether lysosomes could partake in bioPROTAC-mediated degradation, we have now tested two additional lysosome inhibitors: bafilomycin A and chloroquine. Data from degradation and delivery experiments are shown below. We found that chloroquine partially rescued EGFP-KRAS degradation (as did MG-132). While bafilomycin appears to rescue degradation of EGFP-KRAS, a parallel splitGFP complementation assay reveals that no bioPROTAC protein is actually delivered cytosolically. This finding is in agreement with other reports that bafilomycin prevents LNP endosomal escape by inhibiting lysosomal acidification^{5,6}. On the other hand, chloroquine did not block LNP delivery, thus implicating the lysosome in IpaH9.8 bioPROTAC-mediated degradation of EGFP-KRAS. Previous studies have demonstrated chloroquine's ability to inhibit lysosomal degradation without increasing lysosome pH⁷, making it a good candidate for investigating LNP-bioPROTAC degradation mechanisms. Taken together, data from all additional experiments indicate that multiple pathways including proteasomal and lysosomal degradation can contribute to target destruction. We agree with the reviewer that this process is complex. We also believe that these should be investigated on a case-by-case basis, and that observations made in one system are likely not generalizable to all bioPROTAC/target pairs. Specifically,

School of Engineering and Applied Science
Department of Bioengineering
240 Skirkanich Hall
210 South 33rd Street
Philadelphia, PA 19104-6321
Tel 215.898.8501 Fax 215.573.3155

overexpressed EGFP-KRAS appears to degrade via multiple pathways, while endogenous Ras degrades primarily via the UPS when treated with the same bioPROTAC (evidenced by existing supplemental data and additional experiment shown below). Regardless of the degradation mechanism, target depletion still depends on both the activity of the E3 ligase as well as a high-affinity binder. We have included two additional panels in Figure S13 and summarized these findings in the main text.

Table 1. LNP:bioPROTAC degradation assay with inhibitor treatment

[Redacted]

Table 2. LNP:bioPROTAC delivery assay with inhibitor treatment

[Redacted]

School of Engineering and Applied Science
Department of Bioengineering
240 Skirkanich Hall
210 South 33rd Street
Philadelphia, PA 19104-6321
Tel 215.898.8501 Fax 215.573.3155

[Redacted]

Figure 1. Ras degradation in 293T WT cells treated with LNP:bioPROTAC and small-molecule inhibitors.

6. Comments/suggestions following submission of our initial submission: Also, turnover of the bioPROTAC molecule was not investigated – is the bioPROTAC degraded with the target or can it act catalytically? Is the construct turned over without target engagement? With the current format, if the bioPROTAC acts stoichiometrically this could limit the ability to generate and deliver sufficient material to provide a sustained biological effect.

Our initial response to the reviewer's comments: The reviewer asks an important question regarding the fate of our bioPROTACs once delivered into cells. First, we note that the catalytic domain of IpaH9.8 is known to auto-ubiquitinate and contains an auto-inhibitory domain in the LRR domain of the full-length protein (ref 63). This auto-inhibitory domain is likely important to prevent premature degradation of the E3 ligase during host cell immune modulation. We performed further experiments and show that our bioPROTACs auto-ubiquitinate (Figure S12). In addition, fluorescent

School of Engineering and Applied Science
Department of Bioengineering
240 Skirkanich Hall
210 South 33rd Street
Philadelphia, PA 19104-6321
Tel 215.898.8501 Fax 215.573.3155

readout by split GFP is markedly lower in GFP(1-10) reporter cells treated with wild-type IpaH9.8 compared to the C337A variant (catalytically dead) indicating that the protein's catalytic activity leads to its destruction in the cytosol (Figure S13). We suspect that the majority of the auto-degradation is independent of substrate binding, as we observed no difference in the split GFP signal between IpaHWT-K27 and IpaHWT-K27n3. If substrate binding were critical to self-destruction, we might expect that the IpaHWT-K27n3 split GFP signal would be higher, as it is unable to bind to endogenous Ras. Finally, as MG-132 did not rescue GFP complementation, it is possible that auto-ubiquitination leads to degradation through a non-proteasomal pathway such as autophagy.

Ultimately, our bioPROTACs are subject to autoregulation and their therapeutic potencies may be affected by this auto-ubiquitination. Despite this observed self-degradation, we still report rapid and potent degradation of targets in vitro. In future studies, we aim to stabilize this molecule for in vivo applications by identifying and selectively mutating self-ubiquitinated lysine residues, rendering bioPROTACs resistant or immune to auto-ubiquitination. Furthermore, our data indicate that the bioPROTACs are behaving catalytically rather than stoichiometrically. Firstly, delivery of the catalytically-dead IpaH9.8C337A-K27-D25-s11 (100nM) in GFP(1-10) cells produced a fluorescent signal with 58% normalized mean intensity relative to untreated EGFP-KRAS overexpression cells (shown below). This is despite inducing ~95% target degradation. Hence, the protein amount delivered is less than the degradation efficiency, pointing to catalytic activity. We also note that splitGFP is derived from superfolder GFP which is markedly brighter than EGFP (54,000 versus 34,000) making the true difference even larger than what we are noting here. Reconstituted splitGFP has been shown to be as bright as full length superfolder GFP

(Feng, S., Sekine, S., Pessino, V. et al. Improved split fluorescent proteins for endogenous protein labeling. Nat Commun 8, 370 (2017). This allows us to make a reasonable comparison between EGFP (target) and splitGFP (delivery reporter). Further, we are not saturating the pool of GFP(1-10) in 293T reporter cells, as we previously showed even higher complementation levels in the same 293T GFP(1-10) cells with another s11-tagged protein at shorter treatment time lengths than described here (ref 40) In a similar line of reasoning, the total concentration of endogenous Ras in 293T cells has been estimated to be approximately 300nM (ref 77). Our GFP-KRAS construct is overexpressed well above this endogenous level, and treatment with only 100nM of bioPROTAC can almost completely clear the target. Taken together, these data suggest that our bioPROTACs are catalytic, as efficient degradation is achieved at sub-stoichiometric levels.

Comments/suggestions following submission of our revised manuscript: The authors highlight important observations, but the evidence relies on measuring relative levels of fluorescence which could be influenced by a range of factors. Further studies will be

School of Engineering and Applied Science

Department of Bioengineering

240 Skirkanich Hall

210 South 33rd Street

Philadelphia, PA 19104-6321

Tel 215.898.8501 Fax 215.573.3155

required to definitively demonstrate a catalytic mechanism of action; auto-degradation argues against this MOA, experiments to minimise this process would help to support claims of a catalytic process but additional efforts are needed, for example absolute measurements of bioPROTAC and substrate levels. Furthermore, the fact that autophagy may play role again indicates the complexity of the mechanism associated with this construct and that off-pathway perturbations could be significant interfering factors that limit the wider application of the system described.

Response to latest comments/suggestions: Unfortunately, there is no straightforward method that both minimizes autoubiquitination/degradation while simultaneously guaranteeing bioPROTAC activity. As mentioned before, lysine mutagenesis could be attempted, but these efforts would be extensive and offer no promise of retained activity. As such, this strategy is outside the scope of the current work. Any attempts to broadly inhibit native proteasomal/lysosomal activity with small-molecule inhibitors would also block target degradation. We would like to point out that there is no obvious reason why autoubiquitination and catalytic activity should be mutually exclusive. Bearing these in mind, we attempted to quantify whether or not our bioPROTAC catalyzed target degradation in a sub-stoichiometric manner.

To bolster any claims of sub-stoichiometric behavior, we have now measured both the amount of degradation (GFP-KRAS) and delivery (splitGFP complementation) following LNP-mediated delivery. Unlike flow cytometry data presented before, we obtained quantitative values by using a fluorescent protein standard. Degradation was evaluated in 293T GFP-KRAS cells, and delivery was evaluated in 293T GFP(1-10) cells. We used two bioPROTACs: one containing the active E3 ligase and one containing the catalytically-dead C337A variant. At the endpoint (7 hours), treated cells were lysed in 50 μ L non-denaturing lysis buffer, and extracts were measured immediately on a plate reader. The standard curve (linear region shown below) was also made by diluting recombinant eGFP in 293T WT extracts in order to match treatment sample conditions. Degradation and delivery data are presented below. For IpaH9.8(WT) bioPROTACs, the amount of GFP degraded (untreated-treated, orange bars) is much greater than the amount delivered. Anticipating that this could be confounded by self-degradation, we can compare these degradation values to the C337A variant, which does not self-degrade and can be considered an upper limit to the amount of bioPROTAC present in the cytosol. While delivery is higher in this group due to ablated auto-degradation, once again, these values are still below the amount of protein degraded by active bioPROTACs. Using this comparison, target degradation occurs at 2.5X the rate of bioPROTAC delivery, again pointing to sub-stoichiometric behavior.

This assay relies on two key assumptions. Firstly, that our eGFP standard can be used for splitGFP complementation. On this point, we reiterate that splitGFP has already been shown to be as bright as intact sfGFP8, and sfGFP is brighter than eGFP. If anything, our eGFP standard overestimates the amount of protein delivery. Unfortunately, we could not find a vendor for sfGFP. Secondly, we assume that GFP complementation accurately reflects the

School of Engineering and Applied Science

Department of Bioengineering

240 Skirkanich Hall

210 South 33rd Street

Philadelphia, PA 19104-6321

Tel 215.898.8501 Fax 215.573.3155

true amount of delivered protein (i.e., all cytosolic bioPROTAC complexes with GFP(1-10)). The high intracellular GFP(1-10) levels combined with the fact that splitGFP assembly is irreversible should drive free bioPROTAC towards complementation, making this a reasonable assumption to make. To improve this assay, we would need to make absolute measurements of bioPROTAC, but as this is not feasible for us at this time, we rely on the surrogate assay presented here.

We disagree with the reviewer's assertion that a sub-stoichiometric mechanism is required for sustained biological effects. For one, this assumption is not true for any other common therapeutic modality (e.g., small-molecule, antibody). Secondly, these properties can be measured independently of the degradation mechanism. One could simply test the biological/degradation dose-dependent effect of bioPROTAC to determine if enough material is adequately delivered. Here, we report nanomolar degradation efficiency, which is on par with or better than many reported small-molecule PROTACs. This observation is useful in itself regardless of the underlying mechanism. The dose-response is much more informative to the biological outcome, and can be used to benchmark many modalities with varying MOAs against each other. Of course, bioPROTAC stability can impact the degradation efficiency, but it is indeed the degradation efficiency that we and most PROTAC users are interested in. As this current study does not attempt to engineer the protein for increased stabilization, we did not elaborate extensively on the degree of self-degradation.

[Redacted]

Figure 2. Standard curve made by diluting recombinant eGFP in 293T cell lysate/extracts.
School of Engineering and Applied Science
Department of Bioengineering
240 Skirkanich Hall
210 South 33rd Street
Philadelphia, PA 19104-6321
Tel 215.898.8501 Fax 215.573.3155

[Redacted]

Figure 3. Quantitative measurements of bioPROTAC delivery and eGFP degradation. Each data point is the average of n = 2 technical replicates.

To address the reviewer's comment on system complexity, we note that the complexity of ubiquitin-mediated protein processing is not unique to our system, but is instead, intrinsic to ubiquitin PTMs. The fate of ubiquitinated proteins depends on multiple factors including but not limited to: which lysine residues are modified, the types of ubiquitin chains that are propagated, and target protein localization. Notably, AbTAC-mediated degradation of cell surface proteins occurs through ubiquitin-mediated lysosomal processing⁹. More traditional small-molecule PROTACs designed with common VHL- and CRBN-recruiting ligands can also degrade via ubiquitin-mediated lysosome trafficking in addition to UPS^{10,11}. Whereas Lysine-48 linked polyubiquitin chains are known to signal UPS processing, Lysine 63-linked chains directs proteins for autophagy/lysosomal degradation. Consequently, any biodegrader could induce multiple degradation pathways by promoting both types of PTMs¹². Thus, the observation that other pathways other than proteasomal processing may be implicated is not specifically limiting for our system. Instead, such complexity is a feature of TPD writ large, highlighting the value of mechanistic studies for every degradation

School of Engineering and Applied Science
Department of Bioengineering
240 Skirkanich Hall
210 South 33rd Street
Philadelphia, PA 19104-6321
Tel 215.898.8501 Fax 215.573.3155

system. Through additional experiments provided in both rounds of revisions, we have sought to explore such mechanisms, and our data provides insight into context-specific bioPROTAC degradation pathways that have largely not been explored in prior publications (see above response).

7. Comments/suggestions following submission of our initial submission: Regarding the presentation of the manuscript itself, there is a lack of labelling of figures and provision of figure legends. Also, the reporting summary is brief and indicates cells lines were not validated, which is of concern. These elements need more attention.

Our initial response to the reviewer's comments: Figure captions and labels have been added, and all native cell lines used in this study and those used to generate reporter cell lines have been authenticated by STR profiling.

Comments/suggestions following submission of our revised manuscript: Thank you for confirming. In addition, further efforts are required to add information regarding sample sizes to figure legends, this is not clear in all cases, and to make clear how these data are treated in terms of their statistical comparison in the reporting summary, for example whether they represent technical or biological repeats.

Response to latest comments/suggestions: We thank the reviewer for this helpful suggestion. We have specified the replicate type within figure captions and made clarifications to statistical analyses in the reporting summary.

8. Comments/suggestions following submission of our initial submission: Overall, the paper is very interesting but requires additional data to validate the approach, as outlined above. Crucially, further exploration of the biological effect observed when using these tools in relevant pre-clinical models is required—how well do they perform in this context? I expect this will be how most readers/users would want to apply this technology, so a more detailed exploration and validation is essential for maximum impact, ideally with it benchmarked against other delivery systems. Proof-of-concept for in vivo delivery would be transformative.

The authors speculate on future therapeutic potential, but much work needs to be done to understand the efficacy of the constructs described, in addition to their tolerability, before this could even be considered.

Our initial response to the reviewer's comments: We have included three additional studies exploring proteomic changes, inhibitory effects in an in vitro PDAC model, and degradation mechanisms to explore the biological effects of protein bioPROTACs. Moreover, we benchmarked these experiments against bioPROTAC mRNA delivery, a common modality for the field. We believe animal experiments are outside the scope of this study. Additional methods development including targeted

School of Engineering and Applied Science
Department of Bioengineering
240 Skirkanich Hall
210 South 33rd Street
Philadelphia, PA 19104-6321
Tel 215.898.8501 Fax 215.573.3155

protein LNPs and bioPROTAC stabilization might be necessary before undertaking in vivo work and are better suited for a standalone study.

Comments/suggestions following submission of our revised manuscript: Thank you to the authors for carrying out these additional experiments and I agree that in vivo studies are outside of the scope of this study. The new data presented adds to the manuscript and it is exciting to see a protein-based system for the delivery of bioPROTACs. In my view, the manuscript is comprehensive but there remain questions regarding the mechanism of action, particularly how much off-target processes contribute to the anti-proliferative properties observed and the role played by autoubiquitylation and lysosomal degradation. Until the impact of these is better understood it could limit the application of the platform as a tool for proteome editing.

Response to latest comments/suggestions: We are heartened that the reviewer acknowledges our efforts to strengthen the manuscript and finds this work impactful. To further validate our system, we have now included additional follow-up experiments that should clarify lysosomal effects and anti-proliferative effects. In the course of this study, we have revealed a surprising layer of complexity pertaining to bioPROTACs and have worked to address some of these mechanistic questions. Firstly, we used a combination of lysosome/proteasome inhibitors paired with control proteins to tackle questions of degradation mechanisms. By combining bioPROTAC delivery and EGFP-KRAS degradation data, we have revealed context-specific degradation pathways for our bioPROTAC, and these findings could suggest underexplored complexity for the TPD field as a whole. For proliferative studies, we included two additional protein controls, and carefully repeated proliferation experiments to verify that Ras degradation is linked to reduced PDAC growth. However, we disagree that additional proteomics experiments are necessary to support the conclusions of this work. In deciding how to best utilize proteomics resources, we wanted to highlight key differences between bioPROTAC protein delivery compared to other modalities (e.g., mRNA). In response to the reviewer's initial comments, we have already provided additional data regarding self- and target- degradation through inhibitor treatment paired with cloning of mutant IpaH9.8 domains. Like the reviewer, we are also curious about the precise balance of self- and target- degradation. However, at this time, absolute measurements are difficult to obtain and could be addressed more thoroughly in dedicated follow-up studies.

References

School of Engineering and Applied Science
Department of Bioengineering
240 Skirkanich Hall
210 South 33rd Street
Philadelphia, PA 19104-6321
Tel 215.898.8501 Fax 215.573.3155

1. Fleming, J. B., Shen, G. L., Holloway, S. E., Davis, M. & Brekken, R. A. Molecular consequences of silencing mutant K-ras in pancreatic cancer cells: Justification for K-ras-directed therapy. *Molecular Cancer Research* 3, (2005).
2. Patricelli, M. P. et al. Selective inhibition of oncogenic KRAS output with small molecules targeting the inactive state. *Cancer Discov* 6, (2016).
3. Kelly, M. R. et al. Combined proteomic and genetic interaction mapping reveals new ras effector pathways and susceptibilities. *Cancer Discov* 10, (2020).
4. Nolan, A. et al. Proteomic Mapping of the Interactome of KRAS Mutants Identifies New Features of RAS Signalling Networks and the Mechanism of Action of Sotorasib. *Cancers (Basel)* 15, (2023).
5. Hu, B. et al. Thermostable ionizable lipid-like nanoparticle (iLAND) for RNAi treatment of hyperlipidemia. *Sci Adv* 8, (2022).
6. Sahay, G. et al. Efficiency of siRNA delivery by lipid nanoparticles is limited by endocytic recycling. *Nat Biotechnol* 31, (2013).
7. Mauthe, M. et al. Chloroquine inhibits autophagic flux by decreasing autophagosome-lysosome fusion. *Autophagy* 14, (2018).
8. Feng, S. et al. Improved split fluorescent proteins for endogenous protein labeling. *Nat Commun* 8, (2017).
9. Cotton, A. D., Nguyen, D. P., Gramespacher, J. A., Seiple, I. B. & Wells, J. A. Development of Antibody-Based PROTACs for the Degradation of the Cell-Surface Immune Checkpoint Protein PD-L1. *J Am Chem Soc* 143, 593â598 (2021).
10. Qu, X. et al. Effective degradation of EGFR L858R+T790M mutant proteins by CRBN-based PROTACs through both proteasome and autophagy/lysosome degradation systems. *Eur J Med Chem* 218, (2021).
11. Wang, Y. et al. PROTAC-Mediated Selective Degradation of Cytosolic Soluble Epoxide Hydrolase Enhances ER Stress Reduction. *ACS Chem Biol* 18, (2023).
12. Alabi, S. B. & Crews, C. M. Major advances in targeted protein degradation: PROTACs, LYTACs, and MADTACs. *Journal of Biological Chemistry* vol. 296 Preprint at <https://doi.org/10.1016/j.jbc.2021.100647> (2021).

Version 2:

Reviewer comments:

Reviewer #3

(Remarks to the Author)

The authors have gone to a marked effort to address my comments and questions, providing further data on the efficacy of their approach in a disease relevant system and to understand the mechanisms by which this is achieved. Significant additional work will be required to engineer a system that is able to deliver sustained, therapeutically relevant levels of target suppression. The complex mechanisms revealed by these studies will also require further investigation. However, I agree that additional data generation is beyond the scope of this paper, that it will be of interest to the field and represents a novel and interesting approach to delivering bioPROTACs for the targeting of previously 'intractable' proteins.

Given the formatting of the reply, I have attached a separate document which addresses each rebuttal point separately.

Author Rebuttal letter:

REVIEWERS' COMMENTS

Reviewer #3 (Remarks to the Author):

The authors have gone to a marked effort to address my comments and questions, providing further data on the efficacy of their approach in a disease relevant system and to understand the

mechanisms by which this is achieved. Significant additional work will be required to engineer a system that is able to deliver sustained, therapeutically relevant levels of target suppression. The complex mechanisms revealed by these studies will also require further investigation. However, I agree that additional data generation is beyond the scope of this paper, that it will be of interest to the field and represents a novel and interesting approach to delivering bioPROTACs for the targeting of previously intractable proteins.

Given the formatting of the reply, I have attached a separate document which addresses each rebuttal point separately.

Response: We thank the reviewer for their thoughtful feedback on our recent data. We agree with the reviewer that this work is in its early stages, and have added a disclaimer in our discussion section:

Here, we develop such a platform, enabling cytosolic bioPROTAC protein delivery. While our results demonstrate the feasibility of this modality, the technology is still in its early stages, and continued refinement of both the degrader and LNP carrier will be necessary for in vivo therapeutic applications.
